# Recent hemispheric asymmetry in global ocean warming induced by climate change and internal variability

Saurabh Rathore 1,2, Nathaniel L. Bindoff 1,3,4,5✉, Helen E. Phillips 1,5 & Ming Feng 6,7

Recent research shows that 90% of the net global ocean heat gain during 2005–2015 was confined to the southern hemisphere with little corresponding heat gain in the northern hemisphere ocean. We propose that this heating pattern of the ocean is driven by anthropogenic climate change and an asymmetric climate variation between the two hemispheres. This asymmetric variation is found in the pre-industrial control simulations from 11 climate models. While both layers (0–700 m and 700–2000 m) experience steady anthropogenic warming, the 0–700 m layer experiences large internal variability, which primarily drives the observed hemispheric asymmetry of global ocean heat gain in 0–2000 m layer. We infer that the rate of global ocean warming is consistent with the climate simulations for this period. However, the observed hemispheric asymmetry in heat gain can be explained by the Earth's internal climate variability without invoking alternate hypotheses, such as asymmetric aerosol loading.

[1] Institute for Marine and Antarctic Studies, University of Tasmania, Hobart, Australia. [2] ARC Centre of Excellence for Climate System Science, Hobart, Australia. [3] CSIRO Oceans and Atmosphere, Hobart, Australia. [4] Australian Antarctic Program Partnership, Hobart, Australia. [5] ARC Centre of Excellence for Climate Extremes, Hobart, Australia. [6] CSIRO Oceans and Atmosphere, Indian Ocean Marine Research Centre, Crawley, WA, Australia. [7] Centre for Southern Hemisphere Oceans Research, CSIRO, Hobart, Australia. ✉email: n.bindoff@utas.edu.au

Estimates of global ocean heat content (OHC) have improved dramatically since the Argo array obtained global coverage in 2005[1,2]. OHC is the best means available to track the Earth's energy imbalance that is driving ongoing global warming[3]. This energy imbalance is due to the positive radiative forcing of the climate system, which is dominated by the increasing greenhouse gas concentrations, $CO_2$ in particular[4]. More than 90% of the Earth's heat increase is due to this energy imbalance and has been taken up by the ocean[5], as indicated by the long-term trend in OHC[6–8]. The importance of the deep ocean, below the limit of Argo observations (2000 m), has also been highlighted, particularly in efforts to distinguish decadal changes in OHC from long-term trends[9,10].

In the decade of 2001–2012, a hiatus was observed in the increasing long-term trend of globally averaged surface air temperature[11]. This pause in surface warming was the result of trade wind intensification due to the Interdecadal Pacific Oscillation[12,13], and with these changes, there was a corresponding redistribution of energy within the oceans[13–16] and this redistribution is potentially connected to this asymmetric mode. Volcanic events have also contributed to the observed global warming hiatus by increasing the stratospheric loading of sulfate aerosols and cooling the troposphere[17,18]. The hiatus demonstrates the role of internal climate variability and the natural forcing to modify the observed hemispheric rate of heat content change.

Hemispheric asymmetry in global OHC anomaly of the upper 2000 m has been observed during 2005–2015, where the northern hemisphere shows a reduced rate of OHC change, and the southern hemisphere oceans have absorbed 67–98% of the net global ocean heat gain[1,2,19]. The precise cause of this intensified hemispheric asymmetry in OHC is unclear. However, previous studies[1,2,20] suggest that the asymmetric warming may be related to the natural decadal variability or to the high concentrations of aerosols in the northern hemisphere[21], which have contributed to the radiative cooling of the northern hemisphere. Moreover, this asymmetric warming is striking in the presence of large-scale increases in the observational records of the ocean temperatures[22]. This rise in ocean temperature is reflected in the long-term warming of the global ocean, which has shown a rise of OHC in both hemispheres[22,23]. Our study shows that the observed asymmetric ocean warming during 2005–2015 can be explained by the internal climate variability superimposed on the long-term symmetric anthropogenic ocean warming.

## Results

**Temporal variability of OHC anomaly during 2005–2015.** We show the robustness of the hemispheric asymmetry in global OHC change during 2005–2015[1,2,19] using an ensemble of six gridded observational products (see Methods). The depth vs time plot (Fig. 1 and Table 1) of OHC anomaly of the two hemispheres shows the asymmetric character of the upper ocean (0–700 m) with the northern hemisphere cooling and the southern hemisphere warming progressively during 2005–2015. The vertical variations of OHC change in the 0–700 m ocean depth are associated with the ocean dynamics in the Tropical Pacific Ocean related to the El Niño Southern Oscillation (ENSO)[4,24] variability on interannual or longer time scales.

The evolution of the OHC anomaly pattern is also examined in the multi-model mean (MMM) of 11 CMIP5 models for the historical (1980–2005) and RCP 8.5 (2006–2015) simulations (Supplementary Fig. 1). While the observed southern hemisphere trends are consistent with the MMM, this is not the case for the northern hemisphere in 0-700 m. However, the observed OHC change below 700 m clearly shows that both hemispheres have experienced long-term (1980–2015) warming as simulated by the MMM (Supplementary Fig. 1) and unobscured by the internal variability (Fig. 1).

Inspection of Fig. 1 suggests that the observed hemispheric asymmetry of the global OHC in the 0–2000 m depth range is predominantly contributed from the changes in the 0–700 m depth range, which is not present in the MMM (Supplementary Fig. 1). The observed signature of deep ocean warming (700–2000 m) is apparent in both hemispheres and is consistent with the climate model simulations.

**Linear trend in OHC anomaly during 2005–2015.** The rate of heat content change in the global ocean, southern and northern hemisphere for 0–700 m, 700–2000 m, and 0–2000 m depth ranges are shown in Table 1. We find that the net global ocean heat gain in 0–2000 m ($8.38 \pm 0.59 \times 10^{22}$ J decade$^{-1}$) is equally distributed between 0 and 700 m ($4.38 \pm 0.42 \times 10^{22}$ J decade$^{-1}$) and 700-2000 m ($4.00 \pm 0.23 \times 10^{22}$ J decade$^{-1}$), and the southern hemisphere explains around 92% ($7.78 \pm 0.58 \times 10^{22}$ J decade$^{-1}$) of the net global ocean heat gain over 0–2000 m range. In the 0–700 m layer, the southern hemisphere explains 116% of the net global ocean heat gain, and the northern hemisphere's rate of ocean heat gain is negative and offsets southern hemisphere contribution by 16% during 2005–2015. In the 700–2000 m layer, 66% of the net global ocean heat gain is explained by the southern hemisphere and 34% by the northern hemisphere (Table 1).

The linear trends of zonally integrated global OHC from observation (2005–2015) and MMM (2006–2015) are shown in Fig. 2a, b, respectively. While the observed trends represent the combination of internal variability and forced climate change, the MMM will tend to average out internal variability and have a more robust representation of the forced ocean response compared to the observations. Figure 2a shows a robust and enhanced rate of ocean heat gain around 40° S in the observations, which is also in the MMM trend, although weaker in magnitude (Fig. 2b). Cooling and warming patterns around the Equator are also consistent with the MMM trend for this period. In the 0–700 m depth range, observations show the northern hemisphere has a significant reduction in the rate of ocean heat gain around 0°−30° N and north of 40° N (Fig. 2a). This reduced rate of ocean heat gain is not evident in the MMM trend (Fig. 2b) for the same decade. Figure 2a shows that the observed uniform rate of warming in the 700–2000 m depth range in both hemispheres is consistent with the MMM trend (Fig. 2b). This suggests that the ocean below 700 m holds the key to tracking Earth's warming due to climate change since the signal-to-noise ratio there is much higher.

The spatial pattern of observed depth-integrated global OHC anomaly trends in 0–2000 m (Fig. 2c) shows that the southern hemisphere warming is primarily contributed by enhanced ocean heat gain in the subtropical gyres north of 45° S. In the northern hemisphere, the reduction in the rate of OHC change mostly occurs in the tropical and subtropical western Pacific, and the North Atlantic (north of 40° N). The broad-scale observed ocean warming patterns (Fig. 2c) are evident in the MMM trend (Fig. 2d), but the hotspots of enhanced and reduced rates of ocean heat gain are not evident. The striking cooling in 0–2000 m (0–700 m) in the North Atlantic (Fig. 2a and Supplementary Fig. 2) has been described in previous studies[25,26]. The pattern of OHC changes in the 0–2000 m layer (Fig. 2c) primarily reflects the pattern from 0 to 700 m (Supplementary Fig. 2a). The 700–2000 m layer (Supplementary Figs. 1 and 2b) experiences a more uniform pattern of ocean warming in both hemispheres with enhanced warming in the North Atlantic (Supplementary Fig. 2b).

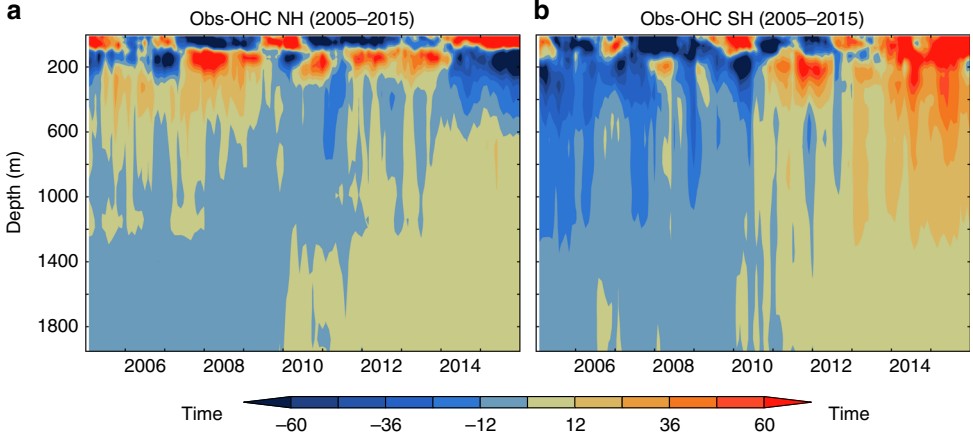

**Fig. 1 Temporal variations of 0–2000 m ocean heat content anomaly.** Hovmöller plot of observed ocean heat content anomaly ($10^{18}$ J m$^{-1}$) in 0–2000 m ocean depth referenced from 2005–2015 for (**a**) northern hemisphere and (**b**) southern hemisphere.

**Table 1 Observed and simulated ocean heat content anomaly trend.**

|  | 0–700 m (MMM) | 700–2000 m (MMM) | 0–2000 (MMM) |
|---|---|---|---|
| Global Ocean | 4.38 ± 0.42 (6.44 ± 1.07) | 4.00 ± 0.23 (2.54 ± 0.53) | 8.38 ± 0.59 (9.0 ± 1.25) |
| Southern Hemisphere | 5.12 ± 0.45 (3.60 ± 0.67) | 2.66 ± 0.19 (1.37 ± 0.39) | 7.78 ± 0.58 (5.0 ± 0.79) |
| Northern Hemisphere | −0.74 ± 0.29 (2.90 ± 0.83) | 1.35 ± 0.14 (1.20 ± 0.41) | 0.60 ± 0.35 (4.0 ± 1.01) |

Ocean heat content anomaly trend ($10^{22}$ J decade$^{-1}$) from the observational mean and multi-model mean (MMM) as the average of RCP 4.5 and RCP 8.5 scenarios (in parentheses) in different depth layers during 2005–2015. Error bars are the 95% confidence interval.

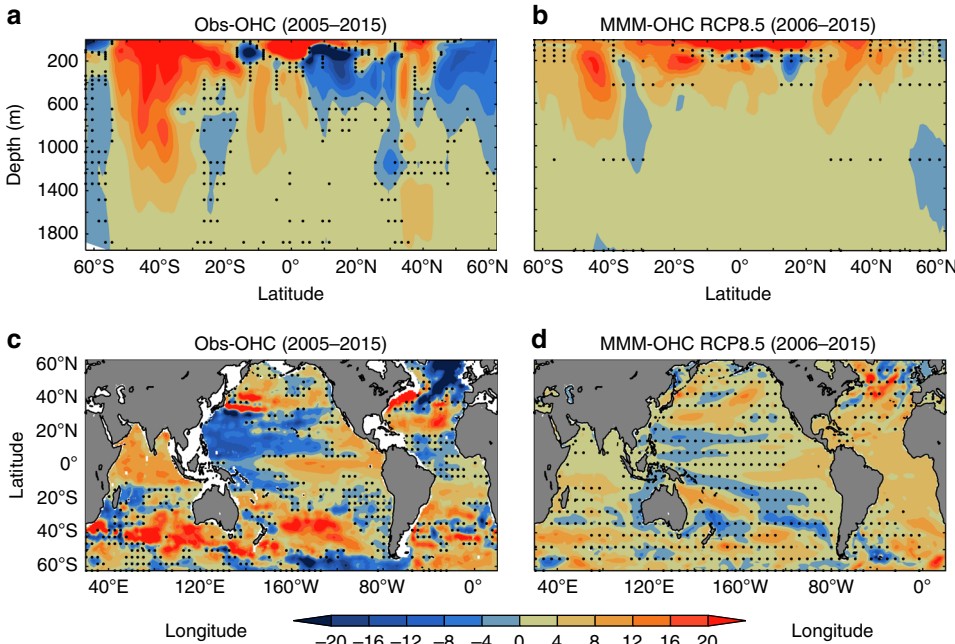

**Fig. 2 Linear temporal trend in ocean heat content anomaly. a** Observed Linear trend for 2005–2015 of zonally integrated global ocean heat content anomaly ($10^{11}$ J m$^{-2}$ year$^{-1}$). **b** Same as (**a**) but for MMM trend for 2006–2015, **c** observed linear trend of global ocean heat content anomaly for 0–2000 m ($10^7$ J m$^{-2}$ year$^{-1}$) for 2005–2015, **d** Same as (**c**) but for MMM trend for 2006–2015. Stippling indicates the locations where OHC anomaly trends are not significant, i.e., <2*standard error of the trends estimated from ($n = 6$) observation products and ($n = 11$) CMIP5 models used in this study.

 

To understand the link between the internal climate variability and hemispheric asymmetry in the global OHC, we use an ensemble of pre-industrial control (Pi-Ctrl), historical (Hist-CMIP5 1980-2005), RCP 4.5 and RCP 8.5 (2006–2015) simulations from 11 CMIP5 models[27] listed in Supplementary Table 1.

**Historical (1980–2005) and RCP (2006–2015) simulated OHC anomaly trends**. The MMM of historical simulations for 0–2000 m (Supplementary Figs. 1 and 3) represents the long-term (1980–2015) externally forced climate change signal in which both hemispheres have warmed symmetrically (Supplementary Fig. 3a, d, g). This symmetrical rate of ocean warming in 0–2000 m depth is not consistent with the observations (Fig. 1 and Table 1). The rate of ocean heat gain in historical simulations (Supplementary Fig. 3) is slightly higher in the southern hemisphere as compared to the northern hemisphere, but both hemispheres show net ocean heat gain. Similarly, the MMM of the RCP 4.5 and RCP 8.5 simulations show similar warming rates in both hemispheres (Supplementary Fig. 3).

The observed OHC anomaly trend in the 700–2000 m depth layer is not obscured by the upper ocean internal variability and represents the long-term warming signature across the globe (Supplementary Fig. 2b) and in both hemispheres (Fig. 1). This long-term warming signature is evident in the historical and RCP simulations (Supplementary Fig. 3c, f, i) but with comparatively higher warming rates. Our analysis shows that the observed asymmetric warming pattern in 0–2000 m (0–700 m) depth is quite distinct from the anthropogenic warming pattern present in the MMM.

It is also interesting to look at the simulated OHC trend pattern for 0–2000 m using the RCP 8.5 scenario from the 11 CMIP5 models (Supplementary Fig. 4). Each of the individual simulations has internal variability with varying phase and a slow climate change signal from anthropogenic forcing superimposed on it. When averaging across all the simulations, the multi-model mean represents the slow underlying climate change signal with the internal variability averaged out. The spatial patterns of the linear trend of 0–2000 m OHC anomaly from individual models (Supplementary Fig. 4) show that the observed warming at 40° S and the cooling signature of the western tropical Pacific along with North Atlantic (as in Fig. 2c), occurs in a few of the individual model simulations (e.g., Supplementary Fig. 4d). The patterns in the MMM (Fig. 2b, d) simulations also suggest that the subduction/ventilation regions on the equatorward side of the western boundary current extensions and Antarctic Circumpolar Current in the Southern Ocean are the hotspots for global ocean heat gain in the 0–2000 m layer.

Unlike the single model realizations (Supplementary Fig. 4), the multi-model mean (Fig. 2d) over this short period of 10-years displays reduced internal variability and evident anthropogenic warming in both hemispheres. This analysis suggests that the observed hemispheric asymmetry of 0–2000 m in the global OHC change (Figs. 1 and 2a, c) is a combination of the internal variability and externally forced anthropogenic warming. However, the strength of the internal variability reduces with the depth, as shown by Fig. 1, exposing a more uniform long-term warming below 700 m.

**Separating internal variability from the forced response**. Using the CMIP5 Pi-Ctrl simulations (Supplementary Table 1), which represent the climate system in the absence of anthropogenic forcing, we investigate whether the observed trend in OHC anomaly during the 11-year Argo period is consistent with internal variability alone. The distribution of internal variability in the Earth system is commonly used in attribution studies[4]. For

this, we select 10-year periods (similar to our observational record length) from the OHC time series of the Pi-Ctrl simulation, which is integrated globally and hemispherically for the depth ranges of 0–2000 m, 0–700 m, and 700–2000 m. We then compute the linear trend over these selected 10-year records of OHC. This procedure is repeated 100,000 times using the Monte-Carlo approach (see Methods for detail).

**OHC anomaly trend for 0–2000 m depth range**. Figure 3a shows the cloud of distribution of 10-year trends of OHC anomaly from the Pi-Ctrl simulations of each model that are concatenated to form the multi-model ensemble (MME, see Methods for detail). This MME represents the internal variability in the northern hemisphere (NH) plotted against southern hemisphere (SH). It shows that the distribution of the 10-year trends in the northern and southern hemispheres tend to be anti-correlated: when the northern hemisphere has a positive rate of change in OHC, the southern hemisphere has negative, and vice versa. Thus, the highest density of points lies in the second and fourth quadrants, and the major axis of this distribution represents an asymmetric internal variability mode. This mode plays a crucial role in the redistribution of ocean heat gain that is internal to the climate system and can exist with or without anthropogenic forcing. It is worth mentioning that there are some instances when the internal variability has in-phase components of heat in both hemispheres and corresponds to the changes in global OHC that implies corresponding changes in the net top-of-atmosphere radiation balance[28]. This in-phase component has a narrower variability compared with the asymmetric internal variations (Fig. 3a). For further investigation, we assume that the characteristics of the internal variability in pre-industrial times will remain the same for the historical simulations (Hist-CMIP5) period of 1980–2005 and the RCP 4.5 and RCP 8.5 simulations of 2006–2015 and beyond. We also consider that the 0–2000 m depth represents the full water column that responds to the internal variations in the net top-of-atmosphere radiation balance[9,28] with the minimum role of vertical ocean heat exchanges.

In contrast to the asymmetric internal variability mode that primarily moves the heat around (without changing the Earth's total energy), the rising GHG concentrations lead to warming of the northern and southern hemisphere oceans in unison (Fig. 3a). This is shown by the linear trend for the recent past from CMIP5 historical simulations (Hist-CMIP5 from 1980–2005) and the linear trend from RCP 4.5 and RCP 8.5 scenarios of the current decade (2005–2016) and for the period of (2020–2100) (Fig. 3a). The MMM of the historical and RCP simulations of CMIP5 models have a positive correlation between northern and southern hemisphere trends, represented by the least square fit line (Fig. 3a). We refer to this line as the direction of climate change, which is approximately normal to the direction of the asymmetric internal variability mode.

In Fig. 3b, we show the evolution in the observed 10-year trends of NH and SH OHC from 1980 to 2016 with a sliding window of 12 months (Hist-Obs). The trends are averages of four observational reanalysis products (ORAS4, ORAS5, SODA3.12.2, and EN4.2.1-G10). This evolution of 10-year trends shows the trajectory of the rate of OHC changes from historical observations and contextualizes the last decade of asymmetric warming of the global ocean. In the early decades, until around 1999 (midpoint of 1995–2004 decade), both hemispheres were warming equally (blue circles, Fig. 3b) like the historical CMIP5 trend (brown diamond, Fig. 3a). Before 1999 the decadal trends lie inside the cloud of Pi-Ctrl (green cloud, Fig. 3b), indicating that the rate of ocean heat gain in both hemispheres was within the range of internal variability.

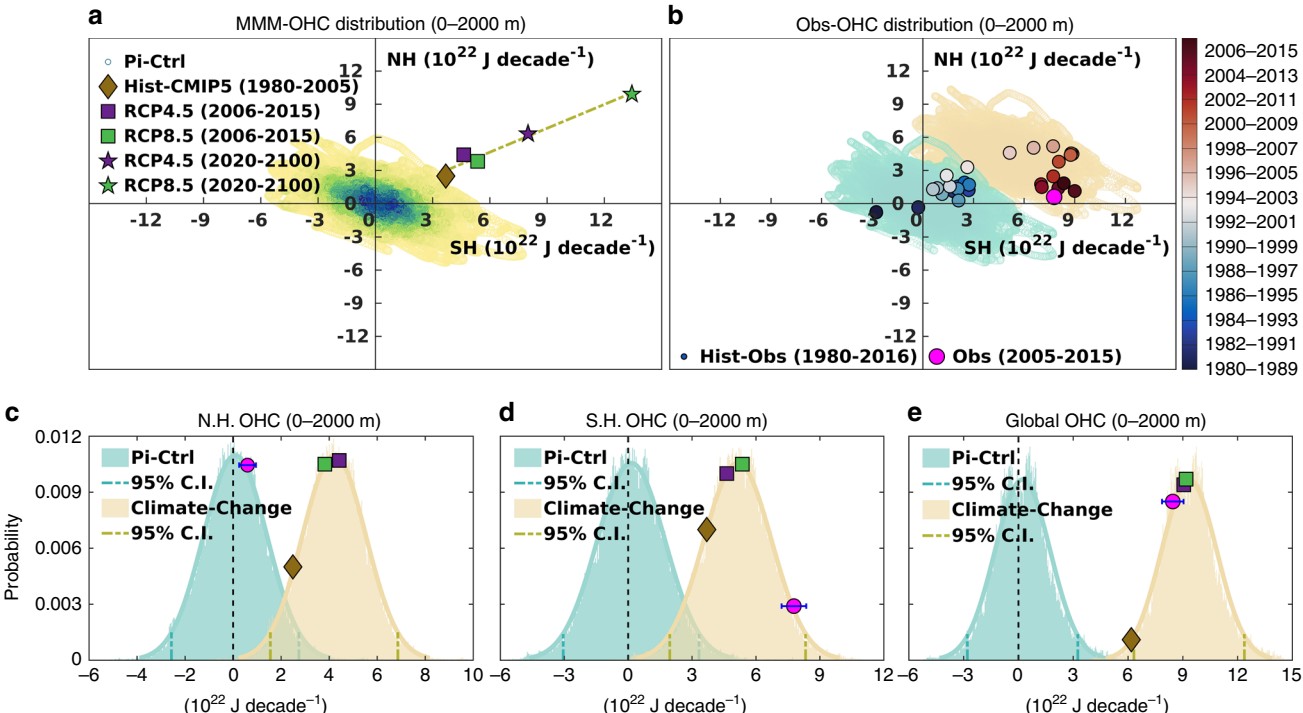

**Fig. 3 Probabilistic analysis of the ocean heat content anomaly trend.** Distribution of the linear trend of the OHC anomaly ($10^{22}$ J decade$^{-1}$) for the depth of 0–2000 m for Northern (NH) and Southern Hemisphere (SH) from (**a**) multi model ensemble (MME) of pre-industrial control (Pi-Ctrl) simulation (cloud), with the multi model mean (MMM) trend from the historical (Hist-CMIP5, brown diamond, 1980–2005), RCP 4.5 (purple square 2006–2015; purple star 2020–2100) and RCP 8.5 (green square 2006–2015; green star 2020–2100) simulation, and the least square fit line passing through these points to represent the direction of climate change (**b**) the green cloud is the same as shown in (**a**) and the orange cloud is the represent the climate change signal in the direction of the best fit line as shown in (**a**), Observed trend over the period of 2005–2015 (Obs, pink circle) along with the trajectory (scatter dots) of the 10-year running trends from the long-term observations over the period of 1980–2016 (Hist-Obs). **c** Probability distribution curve for the northern hemisphere's internal variability (green cloud in (**b**)) and climate change (orange cloud in (**b**)) with the OHC trend from observations (pink circle with the error bar of 95% confidence intervals from two-sided student's *t*-test, 2005–2015), MMM of historical (brown diamond, 1980–2005), RCP 4.5 (purple square, 2006–2015) and RCP 8.5 (green square, 2006–2015) simulations. **d**, **e** Same as (**c**) but for the Southern Hemisphere and the Global Ocean respectively. The 95% confidence interval for the probability distribution curves is derived from the 2-sigma limits for the gaussian distribution of OHC trend.

The onset of hemispheric asymmetry in global OHC begins to appear in the decade of 1996–2005 when the rate of ocean heat gain of the southern hemisphere is faster than the northern hemisphere. All trends (red points, Fig. 3b) since this decade fall outside the range of internal variability (green cloud) from pre-industrial times. This represents the point in time at which robust detection of anthropogenic warming is possible in the context of the Pi-Ctrl estimates of internal variability. The southern hemisphere continued to warm rapidly until the decade 2000–2009, with no further corresponding rate of ocean heat gain in the northern hemisphere. A dramatic decrease in the northern hemisphere's rate of ocean heat gain started from the decade of 1998–2007 (i.e., since 2002, the midpoint of the decade) and has continued to decrease till the last decade of 2007–2016 (dark red points, Fig. 3b). The southern hemisphere's rate of ocean heat gain also reduced from 2001 to 2010 but has increased again recently, and is overshooting the mean trend projected from RCP simulations (Fig. 3a, b).

The asymmetric warming signal has dominated the recent decades with the southern hemisphere absorbing most of the heat gained by the global ocean while the northern hemisphere heat gain is smaller. The observed rate of OHC change in the southern hemisphere during 2005–2015 for the 0–2000 m depth (pink circle, Fig. 3b) is not consistent with the internal variability (green cloud), whereas, the OHC change in the northern hemisphere is entirely in the range of internal variability. Thus, the observed

hemispheric asymmetry during 2005–2015 is an unusual occurrence of the ocean state when compared with the MME of the internal variations (Fig. 3b green cloud) and the MMM of the historical and RCP simulations (Fig. 3a, climate change axis).

Figure 3c–e presents the probability distribution of OHC trends based on the internal variability from the Pi-Ctrl simulations (green cloud) and the Pi-Ctrl plus the average of the MMM trend of RCP 4.5 and RCP 8.5 for 2006-2015 period to represent climate change (orange cloud). The probability distribution for the northern hemisphere's internal variability (green cloud, Fig. 3c) shows the observed rate of northern hemisphere heat gain (pink circle) fits well within the 95% confidence bound of the internal variability. This suggests that it is *very likely* (Probability > 0.90) that the internal variability can account for the observed rate of the northern hemisphere's OHC change. In contrast, the observed rate of ocean heat gain in the southern hemisphere (pink circle, Fig. 3d) exceeds the best-estimated rate of warming from the RCP simulations for the same decade (squares, Fig. 3d). It lies far outside the 95% confidence bound of the southern hemisphere's internal variability (green cloud, Fig. 3d), and at the higher end of the probability distribution with climate change included (orange cloud).

Due to the asymmetric climate mode of internal variability, the reduced rate of ocean heat gain in the northern hemisphere has been compensated by a high heat gain in the southern hemisphere. Most strikingly, the observed warming of the global

ocean (Fig. 3e) is inconsistent with the probability distribution of the internal variability (green cloud) but entirely consistent with the anthropogenic warming (orange cloud) as projected by the RCP simulations from the climate models for the same decade. Note that the magnitude and likelihood of observed warming (pink circle, Fig. 3e) is slightly less than the estimated warming in the climate change scenarios for 2006–2015 but higher than the historical warming trend for 1980–2005 (brown diamond, Fig. 3e).

The observed global OHC trend during 2005–2015 is consistent with the rate of OHC change projected by the RCP simulations and lies outside the cloud of internal variability. However, the contrast between the hemispheric rate of ocean heat gain can be explained by the asymmetrical climate variation. This result provide here is supported by an alternative approach describe in Supplementary Note 2. It is worth mentioning that, based on the CMIP5 multi-model ensemble used in this study, it is *virtually certain* (probability $\cong 0.99$) that the internal variability alone cannot explain the observed warming of the southern hemisphere and the global ocean unless combined with a forced climate change signal (Fig. 3d, e).

This study demonstrates that an anti-symmetric internal variability mode combined with the symmetric pattern of anthropogenic warming in both hemispheres can explain the observed reduction (enhancement) of ocean heat gain in the northern (southern) hemisphere. By taking internal variability correctly into account, we can detect the climate change signal in a short 11-year ocean record as well as account for (in a probabilistic sense) the observed hemispheric asymmetry in global OHC without invoking other forcing mechanisms such as aerosols[2,20].

## Discussion

The probability distribution for the rate of ocean heat gain for both hemispheres in the 0–2000 m depth range (Fig. 3) shows that it is *very likely* that internal variability can explain the observed ocean heat gain of the northern hemisphere in the last decade of 2005–2015. In contrast, it is *virtually certain* that internal variability alone cannot account for the observed warming of the southern hemisphere and the global ocean. Both hemispheres have experienced anthropogenic warming, but our results indicate that the impact of the internal variability for this decade has offset the impact of anthropogenic warming in the northern hemisphere oceans so that there has been no net ocean heat gain in the northern hemisphere (Fig. 3 and Supplementary Fig. 5). In contrast, internal variability has amplified the southern hemisphere warming by shifting heat from the northern hemisphere to the southern hemisphere and adding to the anthropogenic warming there. Detailed investigation of observed hemispheric asymmetry of the global OHC in 0–2000 m (Fig. 3) shows it is primarily confined to the 0–700 m layer (Supplementary Fig. 5). Indeed, it is the combination of anthropogenic climate change and this asymmetric mode of internal variability that provides the physical explanation for the observed enhanced warming of the Southern Ocean[1,2,19,29,30].

The evolution of the observed heat gain (Fig. 3b) shows an unusual decrease in the northern hemisphere and an increase in the southern hemisphere during 2005–2015. Both hemispheres experienced continuous warming in the earlier part of the record and at times, more than the expected anthropogenic rate of ocean heat gain (Fig. 3b). During the year 2000 (midpoint of 1996–2005 decade), OHC trend estimates of the northern (southern) hemisphere shifted to weak (strong) positive rates of ocean heat gain (Fig. 3b and Supplementary Fig. 5b), consistent with the presence of a robust asymmetric internal mode (Fig. 3a and Supplementary

Fig. 5a). This robust asymmetric mode of internal variability results in heat transfer from the northern hemisphere to the southern hemisphere such that more than 90% of the observed net global ocean heat gain has occurred in the southern hemisphere during 2005–2015 (Table 1). Therefore, the pace of southern hemisphere warming exceeds the estimated anthropogenic heat gain from the CMIP5 simulations used here (Fig. 3d).

Furthermore, the depth layer 700–2000 m shows a more robust signature of anthropogenic warming (Supplementary Figs. 1, 2b, and 6), and the noise from internal variability is much weaker than in the surface layer. Our study also shows that the observed rate of ocean heat gain is faster in the 700–2000 m layer compared with the rate projected by climate models (Supplementary Fig. 6). This difference could be due to the lack of resolution of ocean circulation pathways in climate models or subtle errors in the physical parameterizations[28]. Monitoring the deep ocean has distinct advantages for tracking climate change because of weaker internal variability leading to a much higher signal-to-noise ratio. We have shown that the detection of anthropogenic warming is more robust despite the short ocean record used here, but is consistent with earlier approaches for the detection of anthropogenic influence on the climate system[31]. It is striking that the observed global ocean heat gain precisely matches the projected warming in climate model simulations for this decade.

We have noted that the extreme asymmetric case in the recent observation is relatively rare, and it has little effect on the Earth's overall energy balance other than the hemispheric redistribution of the ocean heat gain[13]. Our analysis shows that the observed net heat gain by the global ocean is driven by anthropogenic external forcing of the climate system and that the internal climate variability can explain the hemispheric asymmetry in warming rates. However, the mechanisms that are responsible for generating the anti-correlation of the northern and southern hemisphere warrants further investigation[13,28]. This phenomenon requires substantial changes in net hemispheric air-sea heat exchanges[19] and/or cross-equatorial net ocean heat transport[30,32].

The recognition of the internal variability mode means that the power to detect climate change on shorter periods in the oceans is increased. Moreover, it is not necessary to invoke the other forcing mechanisms[13], such as the asymmetric aerosol loading[2,20] in the atmosphere to explain the hemispheric asymmetry observed in the OHC over the decade of 2005–2015. Despite the high concentration of aerosols in the northern hemisphere that have contributed to its radiative cooling[1,2,20], the combination of the anthropogenic warming and the internal variability of the climate system provides a sufficient and likely explanation for the anomalously enhanced (reduced) rate of ocean heat gain in the southern (northern) hemisphere during 2005–2015.

Our study also emphasizes that the underlying uncertainties can be narrowed down with the understanding of internal variability. This could aid in the closure of energy imbalance[3,7,33] and sea level budgets[6,34–36] with potential improvements in climate models[37,38] to give a better representation of the hemispheric and global changes for the regional and global climate, respectively.

## Methods

**Observation and reanalysis products.** All observational and reanalysis data sets have $1° \times 1°$ spatial resolution, except for SODA3.3.1 ($0.5° \times 0.5°$). Consequently, all the products were re-gridded to a common grid defined by the RG climatology. From the re-gridded observational and reanalysis products, we computed the ensemble mean of the monthly OHC anomalies by using Eq. (1) over the period 2005–2015, calculated as

$$\mathrm{OHC}(x,y,t) = C_p \int_{z_1}^{z_2} \rho(x,y,z,t)\theta(x,y,z,t)\mathrm{d}z \qquad (1)$$

where $C_p$ is the specific heat capacity of seawater 3992 J kg$^{-1}$ K$^{-1}$, $\rho$ is the potential density of seawater computed from potential temperature ($\theta$, degrees K) and practical salinity ($S$, pss) provided by the observations and reanalysis products and depths $z_1$ and $z_2$ (m) are the limits of integration. We calculated OHC for three depth ranges: 0–2000 m, 0–700 m, and 700–2000 m.

**Historical observation from reanalysis products**. For the running trend of 10-year periods from historical observations (Hist-Obs), we used the long-term time series of OHC from 1980–2016 and computed the linear trend in a 10-year window that slides by 12 months. We performed this computation on EN4.2.1, ORAS5, ORAS4, and SODA3.12.2 and then averaged the trends obtained from the four products to get an ensemble mean view.

**CMIP5 products**. To understand the anthropogenic climate change signal and internal variability, we chose 11 CMIP5 models, as shown in Supplementary Table 1. The choice of these 11 CMIP5 models is based on a previous study[30] which shows that the Southern Ocean has high heat content between 40° and 50° S and 11 (out of 12) CMIP5 models robustly capture the pattern of high heat storage on the northern flank of the Antarctic Circumpolar Current, i.e. zonal band of 40°–50° S. The selected[30] CMIP5 models also provide output for the net sea-surface heat flux and thus allow the estimation of ocean heat uptake and ocean heat transport. Moreover, the MMM of the selected models is consistent with the observed trends in the OHC, as shown by our study (Table 1). From these CMIP5 models, we used Pi-Ctrl, historical simulations (Hist-CMIP5, 1980–2005), RCP 4.5, and RCP 8.5 simulations (2006–2015 and 2020–2100) which follow the forcing and experimental design from the CMIP5 protocol[27].

**Remapping and drift correction in CMIP5 models**. Prior to computing the OHC from the CMIP5 models, we remapped the potential temperature (θ) provided by the models from their native grid to 1° × 1° horizontal grid resolution using a bi-linear interpolation scheme. The monthly anomalies of the historical and RCP simulations are computed relative to the base period of 1980–2005 from the historical simulations. To remove the climate drift from the monthly anomalies of θ in each simulation, we subtract the linear trend computed from the monthly time series of the Pi-Ctrl simulation from the same model. The trend was calculated over the full length of the Pi-Ctrl simulation after the seasonal cycle has been removed. There has been a discussion of de-drifting procedures in many studies[28,31,39–42]. However, the approach we have adopted here is the first order linear drift correction that has been recommended[43] to reduce the risk of overfitting. After de-drifting the models, we computed OHC from 11 CMIP5 models used in this study by using Eq. (1) with constant seawater density of (1025 kg m$^{-3}$)[28,29,44] along with the ocean potential temperature (θ) and $C_p$ of 3992 J kg$^{-1}$ K$^{-1}$. We then computed the MMM trend from historical, RCP 4.5, and RCP 8.5 simulations.

**Estimation of internal variability from pre-industrial control simulations**. To quantify the internal variability in the OHC, we used the monthly anomalies of the OHC from the Pi-Ctrl simulations of each CMIP5 model. We conducted an analysis of the global ocean, and separately for the northern and southern hemispheres. Three depth layers were considered for each model: 0–2000 m, 0–700 m, and 700–2000 m. We compute the linear temporal trend over the 10-year periods, a duration similar to our observational record length. For each model, we have used the Monte-Carlo approach for random selection of the 10-year period from the Pi-Ctrl runs and calculate the linear trend over the selected period. We repeat this procedure 100,000 times to generate the synthetic series of 10-year trends. The Monte-Carlo simulations of 10-year periods from all 11 models were then con-catenated into a single series to generate an MME to represent the distribution of all OHC trends due to internal variability (cloud in Fig. 3a). The critical thing to note is that the same 10-year period from the Monte-Carlo simulations was used to estimate trends in the global, northern, and southern hemisphere analyses. Fur-thermore, to represent the climate change due to external forcing, we have shifted the cloud of internal variability (green could in Fig. 3b–e) by the average of the trend estimated from RCP 4.5 and RCP 8.5, i.e., $\left(\frac{RCP4.5_{MMM}+RCP8.5_{MMM}}{2}\right)_{2006-2015}$ which is shown by orange cloud in Fig. 3b–e. The trajectory of historical obser-vation (Hist-Obs 1980–2016, Fig. 3) is computed from the 10-year running trends from the long-term observations over the period of 1980–2016 with a sliding window of 12 months.

**Statistical significance**. We have used the criterion of 2*standard error with a sample size of "n" for the significance testing which is equivalent to the 95% confidence from two-sided student's t-test. The mean trend is significant if it is >2*standard error of the trends estimated from "n" number of observation and CMIP5 models. Here, "n" represents the sample size which is 6 for observational products and 11 for the CMIP5 model used in this study. The confidence intervals for probability distribution curves are derived from the 2-sigma limits for the gaussian distribution of the random variable that corresponds to a 95% confidence interval from a two-sided student's t-test.

## Data availability

For the monthly observational record of 2005–2015, we computed linear trends in OHC from the ensemble mean of six data sets that include two gridded Argo products: Roemmich and Gilson climatology[45] (http://sio-argo.ucsd.edu/) and IPRC Argo (http://apdrc.soest.hawaii.edu/), two sets of objective analysis products of subsurface temperature and salinity from the Hadley Centre[46] (EN.4.2.1) (https://www.metoffice.gov.uk/hadobs/en4/) in which expendable bathythermograph (XBT) biases were corrected in 2009[47] and 2010[48], and two reanalysis products from SODA3.3.1[49] (https://www2.atmos.umd.edu/~ocean/) and ORAS4[50] (https://www.ecmwf.int/en/forecasts/datasets/browse-reanalysis-datasets). To compute the 10-year running trend evolution for historical observations (Hist-Obs) from 1980 to 2016, we used Hadley subsurface analyses EN4.2.1[48], ORAS4[50], ORAS5[51] and SODA3.12.2[49]. The CMIP5 model outputs are available from the Earth System Grid Federation (https://esgf-node.llnl.gov/projects/cmip5).

## Code availability

Data processing codes and the processed data are available from the corresponding author upon request.

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

## Acknowledgements

This research was funded through the Earth System and Climate Change Hub of the Australian government's National Environmental Science Programme. Authors also acknowledge CSHOR, a joint initiative between the Qingdao National Laboratory for Marine Science and Technology (QNLM), CSIRO, University of New South Wales, and University of Tasmania. The assistance of resources from the National Computational Infrastructure supported by the Australian Government and the World Climate Research Programme's Working Group on Coupled Modelling are duly acknowledged for CMIP5 data. Authors thank Rishav Goyal and Ramkrushnbhai Patel for CMIP5 data access and technical help. Authors also want to acknowledge PyFerret and CDO software for analysis and plotting.

## Author contributions

S.R. and N.L.B conceived the study. S.R. performed the analysis in discussion with N.L.B., H.E.P and M.F. All the authors discussed the results and jointly contributed to writing the manuscript.

## Competing interests

The authors declare no competing interests.
