## [Peer Review File · Nature Communications]

Reviewers' comments:

Reviewer #1 (Remarks to the Author):

Review of "Recent hemispheric asymmetry in global ocean warming induced by climate change and internal variability" by S. Rathore, N. L. Bindoff, H.E. Phillips, and M. Feng

This manuscript uses the output from long simulations using 11 coupled climate models to assess possible drivers of the observed trends in interior ocean temperatures over the period 2005-2015, during which the ocean observing system had much more comprehensive coverage than during any previous period. This manuscript finds that the observed hemispheric asymmetry of heating in the upper 700 m can be attributable to a combination of natural modes of variability, but the net warming of the ocean and large rates of heat uptake in the Southern Hemisphere are exceptionally unlikely in the absence of anthropogenic forcing. The techniques and methods used in this paper to analyze these changes are correct and are reasonably well explained. Importantly, this manuscript demonstrates that asymmetric aerosol forcing is not required to drive the observed asymmetry of ocean heat uptake, as has been suggested elsewhere (e.g., by Shi et al., 2018, doi:10.1175/jcli-d-18-0170.1.), and that the observed ocean heat uptake on decadal timescales is broadly consistent with expectations from climate models, once natural variability are taken into account. The attribution of the observed ocean temperature changes between 2005 and 2015 to anthropogenic forcing in this paper is a convincing and important new result. In my view, this manuscript should be published in Nature Communications, but there I also think that substantial improvements to the figures and text of this manuscript before it is accepted would improve the impact of this paper and its accessibility to readers.

Major issues:

1. Clarity of Figures. Figures 3, 4, and S8 are critical for making the main case of this manuscript, but they have substantial problems in their presentation that make them unnecessarily hard to understand.

1a. The text in each of the figures is exceptionally small and difficult to read.

1b. Panel a is very busy, with an apparently color-coded probability distribution of natural variability of northern and southern hemisphere variability from the preindustrial control runs from the climate models, symbols showing historical and projected hemispheric warming averaged over the models for different forcing scenarios, and observed changes over various decadal intervals. These various symbols are very hard to differentiate. These figures currently have 7 panels, but 8 would take the same space. I would recommend splitting panel a) into two – one that includes only the color-code probability distribution of natural variability and a separate panel with a monochrome outline of that distribution (as for the shifted distribution in panel e) overlain with the various symbols for the observed decadal trends and the multi-model mean projections.

1c. One of the factors that would lead to greater heat uptake and variability of the same in the Southern Hemisphere than the northern hemisphere is the simple fact that the Southern Ocean has the much greater ocean volume. It might be worth considering rescaling the axes in panel a) to reflect the different areas of the two hemispheres, so that uniform ocean warming would be along a line with a 45-degree slope.

2. The text seems to have been hastily written in some places, with numerous grammatical errors or missing articles. In particular, there are numerous examples of clauses that start with conjunctions (often "whereas") that are not sentences, but dependent clauses (e.g., lines 114-116; 188-189; 287-288; 306-308). Other sentences have mismatched tense or number. Overall, this manuscript would benefit from more thorough text editing.

Minor points:

3. One important point from this manuscript is that asymmetric forcing is not required to generate the observed asymmetric ocean warming. This point is brought up in the introduction (albeit without any specific references), but it is not raised again in the discussion. I think it should be discussed again after the results are presented.
4. For the benefit of the readership of Nature Communications, greater care should be taken to avoid the use without definition of such jargon as "transient climate sensitivity" (line 41) or "two bias correction" (line 363).
5. The observational analyses are appropriately referenced, but there are no references provided for any of the 11 models that are used in this study. These models are essentially for this study, and they should be properly cited.
6. On line 328, the description of global ocean warming as showing "that the net heat transfer by internal variability ... is negligible" does not many sense to me.
7. The methods sections describe the use of reference density relative to the surface (line 386) or a "constant sea water density" (line 402). This is a small ($\sim 0.5\%$) error, but why make this unnecessary simplification rather than using the actual density of sea-water

Reviewer #2 (Remarks to the Author):

Review of "Recent hemispheric asymmetry in global ocean warming induced by climate change and internal variability" by S. Rathore et al.

The authors show the robust features of ocean heat content (OHC) change in the upper 2000 m of the ocean, noting the strong hemispheric asymmetry, with the Southern Hemisphere accounting for about 90% of the total heat gain during the period 2005-2015. The rest of the manuscript puts the observed large-scale pattern of ocean warming in the context of CMIP5 model simulations of internal variability and the forced climate change response. After conducting analysis that combines model estimates of internal variability and model/observational estimates of the forced climate change response, the authors conclude that the recent observed OHC trends can be explained as a combination of an interhemispheric mode of OHC variability and the forced ocean warming. The study addresses an important topic and the results will no doubt be of wide interest to the climate research community. I find the authors analysis very interesting, appropriate and well-targeted to the specific question of understanding the somewhat puzzling hemispheric asymmetry in recent observed trends in OHC (and by extension, whether these are likely to be indicative of the long-term climate response).

In my view, the main issue for the manuscript is its length/complexity and related to this, the large number of figures, which I think could be reduced substantially. The abstract is concise, and boils down to a few key points: (1) observations show that $> 90\%$ of the OHC gain during in the 0-2000 m for the period 2005-2015 occurred in the Southern Hemisphere; (2) This strong hemispheric asymmetry is NOT consistent the long-term forced response of CMIP5 models, which suggest substantial warming of both hemispheres; (3) However, interhemispheric modes of substantial ocean heat content change are found in CMIP5 model piControl simulations. (4) The observed warming during 2005-2015 can be understood as a superposition of the long-term warming signal and an internally-generated interhemispheric mode of OHC change - it is not necessary to invoke a substantial role for aerosol forcing (although this also cannot be ruled out by the present study).

I would suggest that the manuscript is suitable for publication in Nature Communications subject to major revisions, which I elaborate on below.

Major comments:

1. In my view, there are too many figures in the manuscript, given the relatively simple messages being put across - i.e. 4 figures in the main text with a total of 30 individual panels and another 8 supplementary figures with a total of 91 panels (!).

The authors could reduce this number to make the manuscript easier to follow and less bewildering for the reader. Some specific suggestions would be: (i) focus on the 0-2000 m layer in the main manuscript. The partitioning between 0-700 m and 700-2000 m is an interesting detail that should come out when considering the likely mechanisms; I would further suggest that the depth partitioning in the observations would be better made using a depth-time hovmoller for both N. Hemisphere and S. Hemisphere for the observational period. This would allow you to describe the depth structure without appearing to specify the 0-700 m and 700-2000 m layer partition a priori (and with no real justification in the manuscript, as far as I can see). (ii) particularly in the main manuscript, rather than showing the spatial pattern of trends for every single CMIP5 model, try showing just the multi-model-mean and the multi-model-standard-deviation (or multi-model-range).

2. It is not clear to me what the rotation of the warming signals into "climate change" vs "internal variability" really adds to the analysis/interpretation? I would suggest that the authors remove this step because: (i) it introduces an additional layer of complexity that is not needed to back-up the findings reported in the abstract; (ii) the reader can no longer clearly see the partitioning between the N. and S. Hemisphere when the warming signal and internal variability signal are combined (which seems to be a key issue that the paper is tackling). The probability plots (e.g. Figure 3) could then focus on the 0-2000 m layer for the Globe, N. Hemisphere and S. Hemisphere, reducing Figure 3 from 7 panels to 4 panels (panels b-d of Figure 3 would look similar to panel g).

Specific Minor comments:

3. The text on the 'hiatus' is not very clear. It's clearer if one reserves the term "heat uptake" for global ocean heat content change. This must correspond to changes in Earth's radiative imbalance, since ocean heat storage dominates the global heat budget on annual and longer timescales. I think the argument that needs to be put forward is that a large part of the explanation for the "hiatus" is a vertical re-arrangement of heat in the ocean that occurred primarily in the tropical Pacific.

4. Line 33. "In the past decade" is a bit vague. Please specify the time period in years that you are referring to in this paragraph (e.g. 2008-2018?).

5. The authors should more clearly acknowledge the potential for aerosol forcing in terms of the "hiatus" and the observed OHC trends during 2005-2015. This applies to both anthropogenic and volcanic aerosol forcing.

6. Line 37. Do you really mean that the N Hemisphere oceans are cooler, or simply that they show only a weak signal of warming during the last decade or so?

7. Line 39. Sentence beginning "Previous studies suggest that ..". You need to cite the specific papers that have presented evidence for the role of anthropogenic aerosol forcing. The text here implies a causal relationship between aerosol forcings and TCR, which I think is misleading. The TCR is a property of the model rather than the applied forcing. It is true that climate models with a

strong response to anthro aerosol forcing also have a high climate sensitivity. If retained, this point should be made more clearly. However, I would suggest removing the discussion on TCR here (it could be returned to in a discussion on the implications of the work), since it does not seem directly relevant to the aim of the paper, i.e., to explain the observed trends in OHC.

8. Line 43-44. Please clarify why the hemispheric asymmetry is "surprising". Are the authors suggesting that the recent hemispheric trends may not be representative of the long-term trends (i.e. multi-decadal records of ocean warming that stretch back to the mid-20th century)?

9. Lines 48-55. The use of the term "ocean heat uptake" is potentially confusing here. This term is often used in reference to simple energy balance models of the Earth system, where the *global ocean* acts as a sink term for the radiative imbalance at top-of-atmosphere by absorbing heat and reducing the transient climate change response of surface temperature. The term is clearly meaningful when referring to changes in global ocean heat content (OHC) changes. However, as soon as one moves to smaller domains - either vertical layers or geographic regions, "ocean heat uptake" is no longer meaningful, because observed changes in OHC can occur through both air-sea heat exchange and also vertical or horizontal heat redistribution. In principle, we could see regions of local "heat uptake" and local "heat loss" even in the case where total ocean heat uptake were zero (owing to heat redistribution). Therefore, I would suggest replacing the phrase "ocean heat uptake" with "change in heat storage" or "ocean heat gain" when referring to sub-global domains.

10. Lines 49-51. The authors write "More than 92% of the global ocean heat uptake in the upper 2000 m depth range is equally distributed between 0-700 m and 700-2000 m and primarily absorbed by the southern hemisphere". I don't understand the meaning of this sentence, please clarify. Do you mean that more than 92% of the total heat gain in the 0-2000 m layer is accounted for by warming of the S. Hemisphere, and that there is an equal contribution from the 0-700 m and 700-2000 m layer for this hemispheric warming?

11. Line 87: In the context of the below 700 m layer being more indicative of long-term warming. This can also be inferred from the observations and model simulations by hovmoller plots that show anti-correlated layers in the upper few hundred meters?

12. Figure 1: Please add the multi-model mean panel from Figure S1 and S2 so that a comparison to the "expected" patterns of warming (i.e. multi-model mean) can be made alongside the observations. Please clarify somewhere in the manuscript what the basis of the significance testing is (e.g. is it based on ensemble standard deviation? Is it based on model piControl simulations?). Suggest you use dots to indicate where the signals are NOT significant, since they obscure the very features you are asking the reader to focus on.

13. Lines 64-66: Somewhere in the manuscript the authors should justify why only 11 CMIP5 models were used in the analysis (many more are available).

14. Lines 68-69. The authors assert that the different spatial patterns among the models arise from internal variability, without providing any evidence or explanation for this. The statement could be demonstrated by looking at several ensemble members for a given model, or by characterising the emergent pattern of warming in each model over the 21st century. It is reasonable to suggest that internal variability is the dominant cause of model differences, but some explanation should be given to the reader.

15. Lines 68-78: Regarding the period 1980-2005 to characterise the "long-term underlying climate signal". I think it would be more relevant to choose, e.g., a 30-year period centred on period of the Argo observations - such as 1995-2025. This also has the advantage of avoiding the major volcanic forcing associated with the El Chichon and Pinatubo eruptions of 1982 and 1991. The key issue here seems to be characterization of the "expected warming" under climate change

during the observational period.

16. Line 77-78: The multi-model mean pattern in Figure S1 should be incorporated into Figure 1 in the main manuscript.

17. Lines 131-134 (and elsewhere): I feel that the clarity of the text could be improved here and there, and the authors should take care to say precisely what they mean. In this instance, they refer to "OHC distribution" when what they mean (according to Figure 3) is the distribution of OHC trends, or the OHC tendency.

18. Lines 153-155. I have already commented above that I suggest the authors remove the rotation step from their analysis. Having re-read this, I'm not sure that this decomposition is meaningful (so this may be another reason to remove it). The internal variability also has signature with both hemispheres in phase - and it may be worth pointing out that assuming the 0-2000 m layer is reflective of the full column, this must correspond to internal variations in net top-of-atmosphere radiation (as discussed by Palmer and McNeall, 2014 and other authors) - and therefore cannot be neatly separated from the climate change signal. I think such a separation would require a more sophisticated analysis that would attempt to characterise the spatio-temporal "fingerprints" of the internal variability and forced response.

19. Lines 156-165. As mentioned above, this text would benefit from bringing some physical explanations to things. The in-phase changes of N. and S. Hemisphere OHC change approximately correspond to changes in total Earth system heating via changes in the net radiation balance. The out-of-phase (anti-correlated) changes imply a role for an "internal" (to the climate system) OHC redistribution. I think this also potentially explains why the total variability along this axis is greater - both changes in total Earth system heating and internal heat redistributions can play a role in the hemispheric trends along this axis?

20. Line 194. The authors state: "It is extremely unlikely (probability $\cong 0$) that internal variability alone can explain the observed warming of the SH and for the global ocean". This is a very bold statement on two counts: one it appears to assign a zero probability to something; secondly, it appears to take no account of potential (systematic) deficiencies in the model simulations upon which the statement is premised, and/or the relatively small ensemble size (i.e. 11 models). I think this sentence should be re-phrased to say something like "Based on our model ensemble, internal variability alone cannot explain the observed warming of the SH and global ocean, unless also combined with a forced climate change signal". I would encourage the authors throughout the manuscript to clearly state their assumptions and methodological caveats.

21. Line 342-347. This is very speculative text with no physical insight - it seems to be just a listing of some of the major climate modes. Perhaps the authors should simply state that the mechanisms of this anti-correlation of NH and SH is the subject for future study? In general, the phenomenon must be associated with inter-hemispheric changes in air-sea heat exchange and/or ocean heat transport.

22. Line 348-356. This conclusion is very similar to an earlier study by Roberts et al (2015) in Nature Climate Change - using a similar approach based on piControl simulations. This paper and its findings should be acknowledged somewhere in the text.

Response to Referees

Reviewer #1 (Remarks to the Author):

Review of “Recent hemispheric asymmetry in global ocean warming induced by climate change and internal variability” by S. Rathore, N. L. Bindoff, H.E. Phillips, and M. Feng

This manuscript uses the output from long simulations using 11 coupled climate models to assess possible drivers of the observed trends in interior ocean temperatures over the period 2005-2015, during which the ocean observing system had much more comprehensive coverage than during any previous period. This manuscript finds that the observed hemispheric asymmetry of heating in the upper 700 m can be attributable to a combination of natural modes of variability, but the net warming of the ocean and large rates of heat uptake in the Southern Hemisphere are exceptionally unlikely in the absence of anthropogenic forcing. The techniques and methods used in this paper to analyze these changes are correct and are reasonably well explained. Importantly, this manuscript demonstrates that asymmetric aerosol forcing is not required to drive the observed asymmetry of ocean heat uptake, as has been suggested elsewhere (e.g., by Shi et al., 2018, doi:10.1175/jcli-d-18-0170.1.), and that the

observed ocean heat uptake on decadal timescales is broadly consistent with expectations from climate models, once natural variability are taken into account. The attribution of the observed ocean temperature changes between 2005 and 2015 to anthropogenic forcing in this paper is a convincing and important new result. In my view, this manuscript should be published in Nature Communications, but there I also think that substantial improvements to the figures and text of this manuscript before it is accepted would improve the impact of this paper and its accessibility to readers.

Response:- We thank the reviewer for devoting the time to review this manuscript and providing the encouraging remarks with constructive suggestions.

Major issues:

1. Clarity of Figures. Figures 3, 4, and S8 are critical for making the main case of this manuscript, but they have substantial problems in their presentation that make them unnecessarily hard to understand.

Response:- We have modified figures in our main manuscript as well as in supplementary information in response to reviewer #1. We are now focusing on 0-2000 m depth in our main text which reflects the major contribution from 0-700 m. So now Figure 3 is for the 0-2000 m depth and Supplementary Fig. S5 and S6 is for 0-700 m and 700-2000 m depth respectively. We have also shifted the figure panels and explanation related to the rotation of the coordinate axes into the internal variability and climate change, to the supplementary information Fig. S7. We have proposed the Supplementary Fig. S7 as an alternative method to presenting the climate change signal and internal variability. By shifting this figure from main text to the supplementary information also addresses the complexity issue of representation raised by the reviewer.

1a. The text in each of the figures is exceptionally small and difficult to read.

Response:- We have increased the resolution and image quality throughout the manuscript and also increased the size of the fonts used for axis labels and figure titles.

1b. Panel a is very busy, with an apparently color-coded probability distribution of natural variability of northern and southern hemisphere variability from the preindustrial control runs from the climate models, symbols showing historical and projected hemispheric warming averaged over the models for different forcing scenarios and observed changes over various decadal intervals. These various symbols are very hard to differentiate. These figures currently have 7 panels, but 8 would take the same space. I would recommend splitting panel a) into two – one that includes only the color-code probability distribution of natural variability and a separate panel with a monocolour outline of that distribution (as for the shifted distribution in panel e) overlain with the various symbols for the observed decadal trends and the multi-model mean projections.

Response:- Thank you for your suggestions it was quite helpful for authors also to improve the figures and their information content. We have modified our figure as shown in Fig. 3 and Supplementary Fig. S5, S6 and S7 to be just 5 panels. We have removed the rotated view of the northern and southern hemispheric changes and present these in the Supplementary information (Fig S7) and as suggested above separated out the panels to show the OHC from the models and observations.

1c. One of the factors that would lead to greater heat uptake and variability of the same in the Southern Hemisphere than the northern hemisphere is the simple fact that the Southern Ocean has the much greater ocean volume. It might be worth considering rescaling the axes in panel a) to reflect the different areas of the two hemispheres, so that uniform ocean warming would be along a line with a 45-degree slope.

Response:- Thank you for your suggestion. We did consider re-scaling the axes in earlier versions of the text. The purpose of our choice is to compute the actual rate of heat content change or heat storage within the volume of ocean present in each hemisphere. Thus, maintain the discussion and presentation of the paper in terms of energy.

2. The text seems to have been hastily written in some places, with numerous grammatical errors or missing articles. In particular, there are numerous examples of clauses that start with conjunctions (often “whereas”) that are not sentences, but dependent clauses (e.g., lines 114-116; 188-189; 287-288; 306-308). Other sentences have mismatched tense or number. Overall, this manuscript would benefit from more thorough text editing.

Response:- Thank you for your constructive comment, we have corrected our text and grammatical mistakes throughout the manuscript.

Minor points:

3. One important point from this manuscript is that asymmetric forcing is not required to generate the observed asymmetric ocean warming. This point is brought up in the introduction (albeit without any specific references), but it is not raised again in the discussion. I think it should be discussed again after the results are presented.

Response:- Thank you for your suggestion we have revised the text in the introduction and discussion section as shown below.

Response:- (Introduction, Line 40-43) However, previous studies^{1,2,20} suggest that the asymmetric warming may be related to the high concentrations of anthropogenic aerosols in the northern hemisphere²¹, which have led to a net radiative cooling of the northern hemisphere.

Response:- (Discussion, Line 276-291) We have noted that the extreme asymmetric case in the recent observation is relatively rare, and it has little effect on the Earth’s overall energy balance other than the hemispheric redistribution of the ocean heat gain¹³. Our results also demonstrate that net heat gain by the global ocean is entirely due to anthropogenic warming, and its redistribution is subjected to the internal variability of the climate system. However, the mechanisms that are responsible for generating the anti-correlation of the northern and southern hemisphere warrants further investigation^{13,28}. This phenomenon could be associated with inter-hemispheric changes in air-sea heat exchanges¹⁹ and/or ocean heat transport^{30,32}.

The recognition of the internal variability mode means that the power to detect climate change on shorter periods in the oceans is increased. Moreover, it is not necessary to invoke the other forcing mechanisms¹³, such as the asymmetric aerosol loading^{2,20} in the atmosphere to explain the hemispheric asymmetry observed in the ocean heat content over the decade of 2005-2015. Despite the high concentration of aerosols in the northern hemisphere that indicate a net radiative cooling^{1,2,20}, the combination of the anthropogenic warming and the internal variability of the climate system provides a sufficient and likely explanation for the anomalously enhanced (reduced) rate of ocean heat gain in the southern (northern) hemisphere during 2005-2015.

4. For the benefit of the readership of Nature Communications, greater care should be taken to avoid the use without definition of such jargon as “transient climate sensitivity” (line 41) or “two bias correction” (line 363).

Response:- Thank you for bringing this in our attention. We have removed the description about transient climate sensitivity. We have also modified the text on data availability as shown below.

(Data Availability, Line 365-373), For the monthly observational record of 2005-2015, we computed linear trends in OHC from the ensemble mean of six data sets that include two gridded Argo products: Roemmich and Gilson climatology⁴⁵ (<http://sio-argo.ucsd.edu/>) and IPRC Argo (<http://apdrc.soest.hawaii.edu/>), two sets of objective analysis products of subsurface temperature and salinity from the Hadley Centre⁴⁶ in which expendable bathythermograph (XBT) biases were corrected in 2009⁴⁷ and 2010⁴⁸, and two reanalysis products from SODA3.3.1⁴⁹ and ORAS4⁵⁰. To compute the 10-year running trend evolution for historical observations (Hist-Obs) over the period of 1980-2016, we used Hadley subsurface analyses EN4.2.1⁴⁸, ORAS4⁵⁰, ORAS5⁵¹ and SODA3.12.2⁴⁹.

5. The observational analyses are appropriately referenced, but there are no references provided for any of the 11 models that are used in this study. These models are essential for this study, and they should be properly cited.

Response:- All the references for the CMIP5 model used in our study (along with the modelling centres) are provided in the supplementary information and shown in Supplementary Table (ST1).

6. On line 328, the description of global ocean warming as showing “that the net heat transfer by internal variability ... is negligible” does not many sense to me.

Response:- We have revised our main text for clarity in (Discussion) and also in the supplementary information.

(Discussion, Line 276-281) We have noted that the extreme asymmetric case in the recent observation is relatively rare, and it has little effect on the Earth’s overall energy balance other than the hemispheric redistribution of the ocean heat gain¹³. Our results also demonstrate that net heat gain by the global ocean is entirely due to anthropogenic warming, and its redistribution is subjected to the internal variability of the climate system. However, the mechanisms that are responsible for generating the anti-correlation of the northern and southern hemisphere warrants further investigation^{13,28}.

7. The methods sections describe the use of reference density relative to the surface (line 386) or a “constant sea water density” (line 402). This is a small (~0.5%) error, but why make this unnecessary simplification rather than using the actual density of sea-water.

Response:- (Methods, Line 299-307) In the section of methodology for observations and reanalysis products we computed the potential density " ρ " of sea water by using the potential temperature " θ " and salinity " S " provided with the observations and reanalysis products used in this study.

(Methods, Line 329-343) we have computed OHC from 11 CMIP5 models used in this study by using equation (2) with constant sea water density of (1025 kg/m³)(Gao, Rintoul, & Yu, 2018; Palmer & McNeall, 2014; Palmer et al., 2017) along with the ocean potential temperature (θ) and C_p of 3992 J kg⁻¹ K⁻¹.

However, the method adopted for the computation of potential density in CMIP5 models is well documented by (Gao, Rintoul, & Yu, 2018; Palmer & McNeall, 2014; Palmer et al., 2017) and does not show any major significant difference for consideration.

Reviewer #2 (Remarks to the Author):

Review of “Recent hemispheric asymmetry in global ocean warming induced by climate change and internal variability” by S. Rathore et al.

The authors show the robust features of ocean heat content (OHC) change in the upper 2000 m of the ocean, noting the strong hemispheric asymmetry, with the Southern Hemisphere accounting for about 90% of the total heat gain during the period 2005-2015. The rest of the manuscript puts the observed large-scale pattern of ocean warming in the context of CMIP5 model simulations of internal variability and the forced climate change response. After conducting analysis that combines model estimates of internal variability and model/observational estimates of the forced climate change response, the authors conclude that the recent observed OHC trends can be explained as a combination of an interhemispheric mode of OHC variability and the forced ocean warming. The study addresses an important topic and the results will no doubt be of wide interest to the climate research community. I find the authors analysis very interesting, appropriate and well-targeted to the specific question of

understanding the somewhat puzzling hemispheric asymmetry in recent observed trends in OHC (and by extension, whether these are likely to be indicative of the long-term climate response).

In my view, the main issue for the manuscript is its length/complexity and related to this, the large number of figures, which I think could be reduced substantially. The abstract is concise, and boils down to a few key points: (1) observations show that > 90% of the OHC gain during in the 0-2000 m for the period 2005-2015 occurred in the Southern Hemisphere; (2) This strong hemispheric asymmetry is NOT consistent the long-term forced response of CMIP5 models, which suggest substantial warming of both hemispheres; (3) However, interhemispheric modes of substantial ocean heat content change are found in CMIP5 model piControl simulations. (4) The observed warming during 2005-2015 can be understood as a superposition of the long-term warming signal and an internally-generated interhemispheric mode of OHC change - it is not necessary to invoke a substantial role for aerosol forcing (although this also cannot be ruled out by the present study).

I would suggest that the manuscript is suitable for publication in Nature Communications subject to major revisions, which I elaborate on below.

Response:- We thank the reviewer for appreciating the scope of this study in climate science community. We have tried to address all the concerns through our responses and modifications in the text and figures.

Major comments:

1. In my view, there are too many figures in the manuscript, given the relatively simple messages being put across - i.e. 4 figures in the main text with a total of 30 individual panels and another 8 supplementary figures with a total of 91 panels (!).

The authors could reduce this number to make the manuscript easier to follow and less bewildering for the reader. Some specific suggestions would be: (i) focus on the 0-2000 m layer in the main manuscript. The partitioning between 0-700 m and 700-2000 m is an interesting detail that should come out when considering the likely mechanisms; I would further suggest that the depth partitioning in the observations would be better made using a depth-time hovmoller for both N. Hemisphere and S. Hemisphere for the observational period. This would allow you to describe the depth structure without appearing to specify the 0-700 m and 700-2000 m layer partition a priori (and with no real justification in the manuscript, as far as I can see).

Response:- Thank you for the suggestion. We have added a new figure of Hovmöller diagram (Figure 1) as the response to the reviewer in the result section of the main text under the sub-heading “Temporal variability of ocean heat content from 2005-2015” from Line 52-70.

We have also modified our main manuscript throughout by focusing on 0-2000 m with the required details from 0-700 m and 700-2000 m depth. The figures for the 0-700 m and 700-2000 m depth have now been transferred to Supplementary Information.

The total number of figures in the main text are reduced to 3 with 11 panel and in the supplementary information the total number of figures are reduced from to 7 and with 40 panel.

(ii) particularly in the main manuscript, rather than showing the spatial pattern of trends for every single CMIP5 model, try showing just the multi-model-mean and the multi-model-standard-deviation (or multi-model-range).

Response:- We have shifted this figure to the Supplementary Information as Fig. S4 and we have also included a Whisker plot in Supplementary Fig. S3 which show the range of data used in observation and models with their corresponding statistics. These two steps, as suggested by the reviewer, will address the concern raised about having too many various spatial and zonal section plots from the CMIP5 models.

2. It is not clear to me what the rotation of the warming signals into “climate change” vs “internal variability” really adds to the analysis/interpretation? I would suggest that the authors remove this step because: (i) it introduces an additional layer of complexity that is not needed to back-up the findings reported in the abstract; (ii) the reader can no longer clearly see the partitioning between the N. and S. Hemisphere when the warming signal and internal variability signal are combined (which seems to be a key issue that the paper is tackling). The probability plots (e.g. Figure 3) could then focus on the 0-2000 m layer for the Globe, N. Hemisphere and S. Hemisphere, reducing Figure 3 from 7 panels to 4 panels (panels b-d of Figure 3 would look similar to panel g).

Response:- Thank you, we have taken this suggestion into account along with the one major comment (1) into consideration and revised our Fig. 3 and its description in the main manuscript to focus on 0-2000 m layer. We have transferred the panel about the rotation of the warming signal with its description into the Supplementary Information and proposing it as an alternative method

to understand the internal variability and climate change (Supplementary Information, Fig. S7 with Line 97-133).

Specific Minor comments:

3. The text on the 'hiatus' is not very clear. It's clearer if one reserves the term "heat uptake" for global ocean heat content change. This must correspond to changes in Earth's radiative imbalance, since ocean heat storage dominates the global heat budget on annual and longer timescales. I think the argument that needs to be put forward is that a large part of the explanation for the "hiatus" is a vertical re-arrangement of heat in the ocean that occurred primarily in the tropical Pacific.

Response:- We have modified the manuscript text as shown below and is now clearer.

(Introduction, Line 31-36) In the decade of 2001-2012, a hiatus was observed in the increasing long-term trend of globally-averaged surface air temperature¹¹. This pause in surface warming was the result of trade wind intensification due to the Interdecadal Pacific Oscillation^{12,13} and with these changes there was a corresponding redistribution of energy within the oceans¹³⁻¹⁶. Volcanic events have also contributed to the observed global warming hiatus by increasing the stratospheric loading of sulphate aerosols and cooling the troposphere^{17,18}.

4. Line 33. "In the past decade" is a bit vague. Please specify the time period in years that you are referring to in this paragraph (e.g. 2008-2018?).

Response:- We have now mentioned the specific period of time in the revised manuscript (Introduction, Line 31).

5. The authors should more clearly acknowledge the potential for aerosol forcing in terms of the "hiatus" and the observed OHC trends during 2005-2015. This applies to both anthropogenic and volcanic aerosol forcing.

Response:- We now explain in more detail the role of aerosols (anthropogenic and volcanic) in the observed hiatus and asymmetric pattern of ocean warming in the text of the revised manuscript as suggested by the reviewer and shown below.

(Introduction, Line 34-43) Volcanic events have also contributed to the observed global warming hiatus by increasing the stratospheric loading of sulphate aerosols and cooling the troposphere^{17,18}.

Hemispheric asymmetry in global OHC of the upper 2000 m has also been observed during 2005-2015, where the northern hemisphere shows a reduced rate of OHC change and the southern hemisphere oceans have absorbed 67-98 % of the net global ocean heat gain^{1,2,19}. The precise cause of this intensified hemispheric asymmetry in ocean heat content is unclear. However, previous studies^{1,2,20} suggest that the asymmetric warming may be related to the high concentrations of anthropogenic aerosols in the northern hemisphere²¹, which have led to a net radiative cooling of the northern hemisphere.

6. Line 37. Do you really mean that the N Hemisphere oceans are cooler, or simply that they show only a weak signal of warming during the last decade or so?

Response:- Thank you for highlighting this point. We don't mean that the N. Hemisphere oceans are cooler in the 0-2000 m layer. The 0-700 m layer in the northern hemisphere is cooler and has lost energy. We have modified our text as shown below.

(Introduction, Line 37-39) Hemispheric asymmetry in global OHC of the upper 2000 m has also been observed during 2005-2015, where the northern hemisphere shows a reduced rate of OHC change and the southern hemisphere oceans have absorbed 67-98 % of the net global ocean heat gain^{1,2,19}.

7. Line 39. Sentence beginning "Previous studies suggest that ..". You need to cite the specific papers that have presented evidence for the role of anthropogenic aerosol forcing. The text here implies a causal relationship between aerosol forcings and TCR, which I think is misleading. The TCR is a property of the model rather than the applied forcing. It is true that climate models with a strong response to anthro aerosol forcing also have a high climate sensitivity. If retained, this point should be made more clearly. However, I would suggest removing the discussion on TCR here (it could be returned to in a discussion on the implications of the work), since it does not seem directly relevant to the aim of the paper, i.e., to explain the observed trends in OHC.

Response:- Thank you for your suggestion about TCR. We have revised our text and removed TCR discussion as suggested. The text now reads:

(Introduction, Line 37-43) Hemispheric asymmetry in global OHC of the upper 2000 m has also been observed during 2005-2015, where the northern hemisphere shows a reduced rate of OHC change and the southern hemisphere oceans have absorbed 67-98 % of the net global ocean heat gain^{1,2,19}. The precise cause of this intensified hemispheric asymmetry in ocean heat content is unclear. However, previous studies^{1,2,20} suggest that the asymmetric warming may be related to the high concentrations of anthropogenic aerosols in the northern hemisphere²¹, which have led to a net radiative cooling of the northern hemisphere.

8. Line 43-44. Please clarify why the hemispheric asymmetry is "surprising". Are the authors suggesting that the recent hemispheric trends may not be representative of the long-term trends (i.e. multi-decadal records of ocean warming that stretch back to the mid-20th century)?

Response:- Yes, this scenario of hemispheric asymmetry is quite interesting and surprising on two grounds

(1) (Introduction, Line 43-48) The alternative idea is that this asymmetry is due to natural decadal variability and is striking in the presence of large-scale increases in ocean temperatures in observational records²². The Long-term warming of the global ocean has shown a rise of OHC in both hemispheres on annual to pentadal time scales^{22,23}. Our study demonstrates the link between the observed asymmetric ocean warming during 2005-2015 and internal climate variability on the background of long-term symmetric anthropogenic ocean warming.

(2) The observed rate of warming of both hemispheres is not consistent with the expected rate of warming from CMIP5 multi model mean, which shows near symmetric warming of the global ocean during 2006-2015 in 0-2000 m (0-700 m) depth. The observed northern hemisphere rate of warming undershoots and southern hemisphere rate of warming overshoots the expected rate of warming.

However, the global ocean heat uptake (integrate globally for 0-2000 m) is consistent with the expected rate of warming as projected by the climate models (refer Fig. 3 and Supplementary Fig. S3).

Unlike the depth of 700-2000 m which reflects more the secular trend of anthropogenic warming, the recent hemispheric rate of warming in 0-700 m during 2005-2015 period is not the representative of the expected rate of warming, but the global ocean is, when integrated over the 0-2000 m layer as shown by this study.

9. Lines 48-55. The use of the term “ocean heat uptake” is potentially confusing here. This term is often used in reference to simple energy balance models of the Earth system, where the *global ocean* acts as a sink term for the radiative imbalance at top-of-atmosphere by absorbing heat and reducing the transient climate change response of surface temperature. The term is clearly meaningful when referring to changes in global ocean heat content (OHC) changes. However, as soon as one moves to smaller domains - either vertical layers or geographic regions, “ocean heat uptake” is no longer meaningful, because observed changes in OHC can occur through both air-sea heat exchange and also vertical or horizontal heat redistribution. In principle, we could see regions of local “heat uptake” and local “heat loss” even in the case where total ocean heat uptake were zero (owing to heat redistribution). Therefore, I would suggest replacing the phrase “ocean heat uptake”

with “change in heat storage” or “ocean heat gain” when referring to sub-global domains.

Response:- Thank you for providing the detailed explanation of ocean heat uptake. We have modified the text throughout the manuscript as suggested by the reviewer. We have used the term “ocean heat gain” when referring to the sub global domains either geographically or in vertical depth layers.

10. Lines 49-51. The authors write “More than 92% of the global ocean heat uptake in the upper 2000 m depth range is equally distributed between 0-700 m and 700-2000 m and primarily absorbed by the southern hemisphere”. I don’t understand the meaning of this sentence, please clarify. Do you mean that more than 92% of the total heat gain in the 0-2000 m layer is accounted for by warming of the S. Hemisphere, and that there is an equal contribution from the 0-700 m and 700-2000 m layer for this hemispheric warming?

Response:- We have checked our calculation and have modified the phrase “more than 92%” to “around 92 %”.

Yes, is it correct that we find around 92% of net global ocean heat gain in 0-2000 m is accounted by the increase in heat content in the southern hemisphere (refer Table 1).

Secondly the net global ocean heat gain in 0-2000 m depth is equally distributed between 0-700 m and 700-2000 m depth (refer Table 1).

Based on these comments we have modified our text as shown below.

(Results, Line 72-75) We find that the net global ocean heat gain in 0-2000 m ($8.38 \pm 0.59 \times 10^{22}$ J/decade) is equally distributed between 0-700 m ($4.38 \pm 0.42 \times 10^{22}$ J/decade) and 700-2000 m (4.00

$\pm 0.23 \times 10^{22}$ J/decade), and the southern hemisphere explains around 92% ($7.78 \pm 0.58 \times 10^{22}$ J/decade) of the net global ocean heat gain over 0-2000 m range.

11. Line 87: In the context of the below 700 m layer being more indicative of long-term warming. This can also be inferred from the observations and model simulations by hovmoller plots that show anti-correlated layers in the upper few hundred meters?

Response:- The reviewer's suggestion for the Hovmöller diagram (now Fig. 1 and Supplementary Fig. S1) has been taken into consideration and we have revised our main text as per suggested. Please refer to (Results, Line 61-79) of the result section.

12. Figure 1: Please add the multi-model mean panel from Figure S1 and S2 so that a comparison to the "expected" patterns of warming (i.e. multi-model mean) can be made alongside the observations. Please clarify somewhere in the manuscript what the basis of the significance testing is (e.g. is it based on ensemble standard deviation? Is it based on model piControl simulations?). Suggest you use dots to indicate where the signals are NOT significant, since they obscure the very features you are asking the reader to focus on.

Response:- We have revised Fig. 2 of the main manuscript that shows the comparison of the observed OHC trend (2005-2015) with the expected OHC trend from multi model mean of RCP 8.5 scenario (2006-2015). We also show the comparison of observation with models in Supplementary Fig. S3.

The significance testing of all the linear trends is done by using a two-sided student t-test (Line 358-363). We mention the significance test in the confidence interval of OHC trends in "Table 1" and in the caption of the figures.

We have replaced the dot of Fig. 2 and Supplementary Fig. S2 to indicate the region where the trends are not significant at 95% confidence level from two-sided student's t-test.

13. Lines 64-66: Somewhere in the manuscript the authors should justify why only 11 CMIP5 models were used in the analysis (many more are available).

Response:- Under the section of "Methods" in the main manuscript with the sub-heading "CMIP5 products", we now justified the selection of 11 CMIP5 models used in this study.

(Methods, Line 314-322) To understand the anthropogenic climate change signal and internal variability, we chose 11 CMIP5 models, as shown in Supplementary Table (Table ST1). The choice of these 11 CMIP5 models is based on a previous study³⁰ which shows that the Southern Ocean has high heat content between 40°-50° S. We have used the same 11 (out of 12) CMIP5 models which robustly capture the pattern of high heat storage on the northern flank of the Antarctic Circumpolar Current i.e. zonal band of 40°-50° S. The selected³⁰ CMIP5 models also provide output for the net sea-surface heat flux and thus allow the estimation of ocean heat uptake and ocean heat transport. Moreover, the MMM of the selected models is consistent with the observed trends in the ocean heat content, as shown by our study (Table 1).

14. Lines 68-69. The authors assert that the different spatial patterns among the models arise from internal variability, without providing any evidence or explanation for this. The statement could be

demonstrated by looking at several ensemble members for a given model, or by characterising the emergent pattern of warming in each model over the 21st century. It is reasonable to suggest that internal variability is the dominant cause of model differences, but some explanation should be given to the reader.

Response:- We have rewritten the section “Historical (1980-2005) and RCPs (2006-2015) simulated OHC trends” of the main manuscript from (Result, Line 110-140). Here, we have provided greater context to the spatial patterns and the explanation of internal variability and inter-model differences as shown by Supplementary Fig. S4 with the necessary references (Supplementary information Line 54-61).

15. Lines 68-78: Regarding the period 1980-2005 to characterise the “long-term underlying climate signal”. I think it would be more relevant to choose, e.g., a 30-year period centred on period of the Argo observations - such as 1995-2025. This also has the advantage of avoiding the major volcanic forcing associated with the El Chichon and Pinatubo eruptions of 1982 and 1991. The key issue here seems to be characterization of the “expected warming” under climate change during the observational period.

Response:- Thanks for your suggestion. We chose to have the reference period of 1980-2005 to characterise the “long-term underlying climate signal” and because this is a reasonably well observed period before the Argo era and is consistent with the periods used by IPCC AR5.

16. Line 77-78: The multi-model mean pattern in Figure S1 should be incorporated into Figure 1 in the main manuscript.

Response:- We have provided our response in the 12th comment of the reviewer. See the revised Fig. 2 of the main manuscript and Supplementary Fig. S3 from the Supplementary Information.

17. Lines 131-134 (and elsewhere): I feel that the clarity of the text could be improved here and there, and the authors should take care to say precisely what they mean. In this instance, they refer to “OHC distribution” when what they mean (according to Figure 3) is the distribution of OHC trends, or the OHC tendency.

Response:- We have modified the text as shown below.

(Results, Line 151-158) Fig. 3a shows the cloud of distribution of 10-year trends of OHC from the pre-industrial control (Pi-ctrl) simulations of each model that are concatenated to form the multi-model ensemble (MME, see methods for detail). This MME represents the internal variability in the northern hemisphere (NH) plotted against southern hemisphere (SH). It shows that the distribution of the 10-year trends in the northern and southern hemispheres tend to be anti-correlated: when the northern hemisphere has a positive rate of change in OHC, the southern hemisphere has negative, and vice versa. Thus, the highest density of points lies in the 2nd and 4th quadrants, and the major axis of this distribution represents an asymmetric internal variability mode.

18. Lines 153-155. I have already commented above that I suggest the authors remove the rotation step from there analysis. Having re-read this, I’m not sure that this decomposition is meaningful (so

this may be another reason to remove it). The internal variability also has signature with both hemispheres in phase - and it may be worth pointing out that assuming the 0-2000 m layer is reflective of the full column, this must correspond to internal variations in net top-of-atmosphere radiation (as discussed by Palmer and McNeall, 2014 and other authors) - and therefore cannot be neatly separated from the climate change signal. I think such a separation would require a more sophisticated analysis that would attempt to characterise the spatio-temporal “fingerprints” of the internal variability and forced response.

Response:- The suggestion by the reviewer has been considered by the authors and Fig. 3 in the main manuscript and Supplementary Fig. S5 and S6 of the Supplementary Information has been modified.

However, we have shown the rotation step in the Supplementary Information in Supplementary Fig. S7 as an alternate approach to detect the climate change signal.

We have provided the explanation for the distribution of 10-year trends from the multi model ensemble of the Pi-Control simulation under the sub-heading “Separating internal variability from climate change in the observed OHC trend” from (Results, Line 151-167) along with the assumptions related to the internal variability and ocean water column.

19. Lines 156-165. As mentioned above, this text would benefit from bringing some physical explanations to things. The in-phase changes of N. and S. Hemisphere OHC change approximately correspond to changes in total Earth system heating via changes in the net radiation balance. The out-of-phase (anti-correlated) changes imply a role for an “internal” (to the climate system) OHC redistribution. I think this also potentially explains why the total variability along this axis is greater - both changes in total Earth system heating and internal heat redistributions can play a role in the hemispheric trends along this axis?

Response:- We have provided the physical explanation of the internal variability and its distribution with necessary assumptions as suggested by the reviewer in (Results, Line 151-167) under the sub-heading “Separating internal variability from climate change in the observed OHC trend” and (Discussion, Line 276-283) under “Discussion” section.

20. Line 194. The authors state: “It is extremely unlikely (probability \cong 0) that internal variability alone can explain the observed warming of the SH and for the global ocean”. This is a very bold statement on two counts: one it appears to assign a zero probability to something; secondly, it appears takes no account of potential (systematic) deficiencies in the model simulations upon which the statement is premised, and/or the relatively small ensemble size (i.e. 11 models). I think this sentence should be re-phrased to say something like “Based on our model ensemble, internal variability alone cannot explain the observed warming of the SH and global ocean, unless also combined with a forced climate change signal”. I would encourage the authors throughout the manuscript to clearly state their assumptions and methodological caveats.

Response:- We have rephrased our text throughout the manuscript with better assumptions and explanation as suggested by the reviewer. The response is shown below.

(Results, Line 207-213) The probability distribution for the northern hemisphere’s internal variability (green cloud, Fig. 3c) shows the observed rate of northern hemisphere heat gain (pink circle) fits well

within the 95% confidence bound of the internal variability. This suggests that it is very likely (Probability > 0.75) that the internal variability can account for the observed rate of the northern hemisphere's OHC change. In contrast, the observed rate of ocean heat gain in the southern hemisphere (pink circle, Fig. 3d) exceeds the best-estimated rate of warming from the RCP simulations for the same decade (squares, Fig. 3d).

(Line 227-231) It is worth mentioning that, based on the CMIP5 multi model ensemble used in this study, it is virtually certain that the internal variability alone cannot explain the observed warming of the southern hemisphere and the global ocean unless combined with a forced climate change signal (Fig. 3d and 3e).

21. Line 342-347. This is very speculative text with no physical insight - it seems to be just a listing of some of the major climate modes. Perhaps the authors should simply state that the mechanisms this anti-correlation of NH and SH is the subject for future study? In general, the phenomenon must be associated with inter-hemispheric changes in air-sea heat exchange and/or ocean heat transport.

Response:- We have modified our text to include physical insight and provided the references for the associated mechanisms.

(Discussion, Line 276-283) We have noted that the extreme asymmetric case in the recent observation is relatively rare, and it has little effect on the Earth's overall energy balance other than the hemispheric redistribution of the ocean heat gain¹³. Our results also demonstrate that net heat gain by the global ocean is entirely due to anthropogenic warming, and its redistribution is subjected to the internal variability of the climate system. However, the mechanisms that are responsible for generating the anti-correlation of the northern and southern hemisphere warrants further investigation^{13,28}. This phenomenon could be associated with inter-hemispheric changes in air-sea heat exchanges¹⁹ and/or ocean heat transport^{30,32}.

22. Line 348-356. This conclusion is very similar to an earlier study by Roberts et al (2015) in Nature Climate Change - using a similar approach based on piControl simulations. This paper and its findings should be acknowledged somewhere in the text.

Response:- We have duly acknowledge the work from Roberts et al (2015) in the "Introduction" in (Introduction, Line 32-34) "Discussion" section in (Discussion, Line 276-281).

REVIEWERS' COMMENTS:

Reviewer #1 (Remarks to the Author):

Review of the revised version of “Recent hemispheric asymmetry in global ocean warming induced by climate change and internal variability” by S. Rathore, N. L. Bindoff, H.E. Phillips, and M. Feng

This revised manuscript is a great improvement over its predecessor, addressing all of major issues that I (and the other reviewer) raised in my previous review. The key figures are now much clearer, and the manuscript now makes a clear and compelling argument that the observed 2005-2015 hemispherically integrated 0-2000 m ocean warming patterns are consistent with an anthropogenically forced warming signal along with a mode of natural variability that are both consistent with coupled climate model simulations. There are a few additional minor changes to the text that I would like to suggest, after which I think that this manuscript should be published in in Nature Communications.

Minor issues:

1. Line 77: The phrase “the northern hemisphere’s rate of ocean heat gain is reduced by 16%” does not make sense and could easily be misinterpreted. This clause makes it sound like the northern hemisphere is warming at 84% of the global mean rate. Instead, I would say that “In the 0-700 m layer, integrated northern hemisphere cooling offsets 14% ($=0.74/5.12$) of the southern hemisphere warming during 2005-2015.”
2. Line 104: The phrase “... more enhanced warming in the North Atlantic” is confusing to me. I think that would make more sense if it simply read “...enhanced warming in the North Atlantic”.
3. Lines 143, 147, 152, 209, 326, and 354. The shorthand notation “Pi-control” is used in place of “pre-industrial control” without ever being defined as such. Elsewhere in the text, “Pi-ctrl” is explicitly defined as shorthand for “pre-industrial control”. To avoid confusion for the readers, consistent notation should be used throughout the manuscript and explicitly defined at its first use.
4. Line 288: The phrase “the current ocean state is anomalous to the expected internal variations” is confusing and should be clarified.
5. Line 231-232: If there is a specific statistical meaning that can be given to the phrase “virtually certain”, such as “virtually certain (less than 1% probability)”, that should be added.

Reviewer #2 (Remarks to the Author):

General Comments:

The author team has substantially improved the manuscript since the initial submission and I am generally satisfied with their response to my previous review comments. I recommend that the work

is published subject to the authors addressing my comments below, which mostly concern improving the clarity/presentation of the work.

I have not looked extensively through the Supplementary Materials due to time constraints, and I would encourage the author team to look carefully again through that section before submitting a final version of the manuscript. I think this is a very well-conceived and important piece of research and hence my enthusiasm for helping to “polish” the final presentation.

Detailed Comments:

1. Abstract. I have the following suggestions that I feel tighten up the abstract text a bit. My suggested text reads as follows:

“Recent research shows that 90% of the net global ocean heat gain during 2005-2015 was confined to the southern hemisphere with little corresponding heat gain in the northern hemisphere ocean. We propose that this heating pattern of the ocean is driven by anthropogenic climate change and an asymmetric climate variation. This asymmetric variation is found in the pre-industrial control simulations from 11 climate models. While both layers (0-700 m and 700-2000 m) experience steady anthropogenic warming, the 0-700 m layer experiences larger internal variability, which primarily drives the observed hemispheric asymmetry of global ocean heat gain in 0-2000 m layer. We demonstrate that the rate of global ocean warming is consistent with the climate model simulations for this period. Moreover, the observed hemispheric asymmetry in heat gain can be explained by the Earth’s internal climate variability without invoking alternate hypotheses, such as asymmetric aerosol loading.”

2. Lines 25-27. This sentence is confusing because aerosol forcing of the climate system would lead to loss of energy, not a gain. The gain of energy must be due to positive radiative forcing of the climate system, which as I’m sure the authors know, is dominated by greenhouse gases (and CO₂ in particular).

3. Lines 31-36. It is not clear what this section on the “hiatus” has to do with the rest of the manuscript. Suggest that you either make it clear this is a demonstration of the ability of (unforced?) ocean heat redistribution to modify the observed rates of heat content change, or consider removing this paragraph.

4. Line 37. Remove the word “also”.

5. Line 42. I would challenge the sentence “which have led to a net radiative cooling of the northern hemisphere”. Is this really true? This implies that the N Hemisphere *lost* heat during this period (this isn’t true of OHC change?). A more robust statement might be “which have contributed to radiative cooling of the northern hemisphere”.

6. Line 43. Radiative forcing and internal variability as hypotheses are not mutually exclusive – both may have played a role. In addition, we cannot rule out the possibility that aerosol (or other) forcings

may have interacted in some way with internal variability.

7. Lines 44-46. “annual to pentadal time scales” seems to contradict “long-term warming”. Please clarify this point.

8. Lines 46-48. Here and elsewhere. I think that “demonstrate” is too strong a word and implies a greater level of certainty about the real world than we can have based on a small ensemble of climate model simulations. Here’s a suggested revised sentence: “Our study shows that the observed asymmetric ocean warming during 2005-2015 can be explained by internal climate variability superimposed on the long-term symmetric anthropogenic ocean warming”.

9. Line 51. Replace “robust” with “robustness of the”.

10. Line 54 / Figure 1. The reader cannot really tell whether the Northern Hemisphere shows a net cooling or not. This requires one to “integrate by eye” over a plot in which the colorscale is saturated. Could you add a line to show the net OHC change, perhaps with a second y-axis?

11. Line 55-58. Here and elsewhere – please avoid subjective terms like “shallower”, “shallow”, “deep”, “long-term” etc. Rather state the depth or time-horizons you are referring to. Based on this plot, we cannot infer the relative importance of air-sea heat fluxes vs dynamical processes. I think the key point is that we know that a large part of the anti-correlated layers of 0-100m and 100-600m are associated with ocean dynamics in the Tropical Pacific related to ENSO variability. Suggest revising the text to emphasise this point? Final sentence – based on the plot, this variability can be longer than year-to-year (looking at temporal characteristics of the anti-correlated layers)?

12. Figure 1 and Figure S1 (and elsewhere?): Please specify the “baseline” period for the Hovmoller plots (i.e. the reference period you compute the OHC anomaly relative to). In both captions – you are not showing “observed ocean heat content”. You are showing “observed ocean heat content anomaly” [heat content would always be positive].

13. Figure S1. Please format this figure in exactly the same way as Figure 1 in the main manuscript to aid the comparison you are inviting the reader to make in lines 59-64.

14. Lines 61-62. Replace “well represented in” with “are similar to”.

15. Lines 65-66. Replace “Analysis” with “Inspection” and replace “variability” with “changes”.

16. Lines 67-70. Replace “well represented by” with “consistent with”.

17. Section “Linear trend in ocean heat content during 2005-2015”. I think it would be helpful to explain to the reader that while the observed trends represent the combination of internal variability and forced climate change, the MMM will tend to average out internal variability and have a stronger representation of the forced ocean response compared to the observations.

17. Lines 75-77. “.. reduced by 16% during 2005-2015”. Reduced compared to what? Or do you

mean that the heat gain is negative and constitutes -16% of the total heat gain in that layer? Please clarify.

18. Table 1. I think it would be helpful to include the warming rates from the MMM as well. A key point would be the extent to which the 0-2000 m global ocean rates match between models and observations (one could indicate the MMM spread as part of this)?

19. Figure 2. I don't understand what the methodologies are for the significance testing – this should be made clear somewhere in the manuscript or the supplementary materials. Once the authors have clarified the methods they should make clear what the interpretation of these results for the reader in the main text. An alternative would be to remove the significance testing from the figure. This would allow the authors to focus their discussion on the *differences* in the observed and MMM trends in OHC. This seems to be the key point of the figure? Relative to the MMM, the observations show higher warming rates in the Southern Ocean and some large areas of cooling in the N. Hemisphere.

20. Lines 83-93 and elsewhere. I would suggest that the authors use a different term than “well represented” when referring to the MMM simulations compared to the observations, e.g., “consistent” or similar. The thrust of the paper argues that internal variability has played a large role in shaping the observed changes, and we know that the MMM is designed to remove this internal variability – so the idea that observations are “well represented” seems a slightly confusing message.

21. Lines 110-111. Replace “underlying climate change signal” with “externally forced climate change signal”.

22. Lines 116-117. Replace “symmetric warming patterns” with “similar warming rates”.

23. Line 118-119. Please cite a figure and make it clear that you are referring to a spatially more uniform pattern of warming. I believe the authors are referring to Figure S2b.

24. Lines 124-128 and Figure S4. To help the reader, please introduce a final panel to this figure that shows the MMM result (there is a space for this bottom right).

25. Line 137-139. Replace “analysis shows” with “analysis suggests” and insert “externally forced” before “anthropogenic warming”.

26. Line 141. Please replace “from climate change” with “from forced response” or similar. Climate change can arise through both internal variability or external forcings, especially given the relatively short time-horizons being discussed in this paper.

27. Line 142. Replace “the worlds without” with “representations of the climate system in the absence of”.

28. Line 145. Remove the word “event”.

29. Line 161-162. Replace “the Earth’s total energy due to changes in the net radiation balance” with “global ocean heat content that implies corresponding changes in the net top-of-atmosphere radiation balance”.

30. Line 167. In relation to comment above – either change to “top-of-atmosphere” here or amend comment above. Either way, best to use consistent naming convention throughout the manuscript.

31. Figure 3. The caption is very long. I suggest the authors try to reduce this by putting some of this information in the methods section.

32. Lines 271-272. Suggest hyphenating “signal-to-noise”, here and elsewhere.

33. Line 278-280. “Entirely due” is a categorical statement, which is unnecessary, and open to criticism. Suggest re-phrase along the lines of “model analysis shows that the observed net heat gain by the global ocean is driven by anthropogenic external forcing of the climate system and that the hemispheric asymmetry in warming rates can be explained by internal climate variability”.

34. Line 281-283. Suggest the text is replaced with: “This phenomenon requires substantial changes in net hemispheric air-sea heat exchanges and/or cross-equatorial net ocean heat transport”. This statement of course assumes a minimal role for vertical ocean heat exchange – it may be worth stating this somewhere explicitly in the paper (perhaps here?) since I believe it is a central assumption to the analysis?

35. Line 288. Again, I question the evidence for “net radiative cooling” – see comment 5.

36. Method section. Equation (1) and (2) appear to be identical? Please remove the second occurrence.

37. Lines 315-318. This sentence is not clear. Are you saying that the models were chosen because they have a better representation of the S. Ocean mean state? (i.e. relatively smaller biases?). I think you can probably merge these sentences.

38. Line 324. “climate forcing” is not observed. I would simply state that the forcings and experimental design follow the CMIP5 protocol. Please include references to CMIP5 and the experimental documentation here (I believe that is Taylor et al, 2012 for CMIP5, and Meinshausen et al, 2011, for RCPs. I’m not sure what papers describe “historical” and “pre-industrial” experiments).

Reviewers' comments:

Reviewer #1 (Remarks to the Author):

Review of the revised version of "Recent hemispheric asymmetry in global ocean warming induced by climate change and internal variability" by S. Rathore, N. L. Bindoff, H.E. Phillips, and M. Feng

This revised manuscript is a great improvement over its predecessor, addressing all of major issues that I (and the other reviewer) raised in my previous review. The key figures are now much clearer, and the manuscript now makes a clear and compelling argument that the observed 2005-2015 hemispherically integrated 0-2000 m ocean warming patterns are consistent with an anthropogenically forced warming signal along with a mode of natural variability that are both consistent with coupled climate model simulations. There are a few additional minor changes to the text that I would like to suggest, after which I think that this manuscript should be published in Nature Communications.

Response:- We thank the reviewer for devoting the time to review the manuscript and providing the encouraging remarks with constructive suggestions for publication in Nature Communications.

Minor issues:

1. Line 77: The phrase "the northern hemisphere's rate of ocean heat gain is reduced by 16%" does not make sense and could easily be misinterpreted. This clause makes it sound like the northern hemisphere is warming at 84% of the global mean rate. Instead, I would say that "In the 0-700 m layer, integrated northern hemisphere cooling offsets 14% ($=0.74/5.12$) of the southern hemisphere warming during 2005-2015."

Response:- We have modified the statement.

(Result, Line 76-79) In the 0-700 m layer, the southern hemisphere explains 116% of the net global ocean heat gain, and the northern hemisphere's rate of ocean heat gain is negative and offsets southern hemisphere contribution by 16% during 2005-2015.

2. Line 104: The phrase "... more enhanced warming in the North Atlantic" is confusing to me. I think that would make more sense if it simply read "...enhanced warming in the North Atlantic".

Response:- We have made the correction as suggested by the reviewer.

(Results, Line 102-104) The 700-2000 m layer (Supplementary Figure 1 and 2b) experiences a more uniform pattern of ocean warming in both hemispheres with enhanced warming in the North Atlantic (Supplementary Figure 2b).

3. Lines 143, 147, 152, 209, 326, and 354. The shorthand notation "Pi-control" is used in place of "pre-industrial control" without ever being defined as such. Elsewhere in the text, "Pi-ctrl" is explicitly defined as shorthand for "pre-industrial control". To avoid confusion for the readers, consistent notation should be used throughout the manuscript and explicitly defined at its first use.

Response:- We have made the correction for the abbreviations used in the manuscript as suggested by the reviewer. On the first use of “pre-industrial control” in (Line 106), we have used “Pi-Ctrl” onwards.

4. Line 288: The phrase “the current ocean state is anomalous to the expected internal variations” is confusing and should be clarified.

Response:- We have modified the statement with the additional explanation for better understanding.

(Results, Line 226-229) The observed global ocean heat content trend during 2005-2015 is consistent with the rate of ocean heat content change projected by the RCP simulations and lies outside the cloud of internal variability. However, the contrast between the hemispheric rate of ocean heat gain can be explained by the asymmetrical climate variation.

5. Line 231-232: If there is a specific statistical meaning that can be given to the phrase “virtually certain”, such as “virtually certain (less than 1% probability)”, that should be added.

Response:- We have mentioned the phrase “virtually certain” as “probability $\cong 0.99$ ”.

(Results, Line 230-233) It is worth mentioning that, based on the CMIP5 multi-model ensemble used in this study, it is virtually certain (probability $\cong 0.99$) that the internal variability alone cannot explain the observed warming of the southern hemisphere and the global ocean unless combined with a forced climate change signal (Fig. 3d and 3e).

Reviewer #2 (Remarks to the Author):

General Comments:

The author team has substantially improved the manuscript since the initial submission and I am generally satisfied with their response to my previous review comments. I recommend that the work is published subject to the authors addressing my comments below, which mostly concern improving the clarity/presentation of the work.

I have not looked extensively through the Supplementary Materials due to time constraints, and I would encourage the author team to look carefully again through that section before submitting a final version of the manuscript. I think this is a very well-conceived and important piece of research and hence my enthusiasm for helping to “polish” the final presentation.

Response:- We thank the reviewer for appreciating the revised manuscript and providing the suggestions for better representation of the findings.

Detailed Comments:

1. Abstract. I have the following suggestions that I feel tighten up the abstract text a bit. My suggested text reads as follows:

“Recent research shows that 90% of the net global ocean heat gain during 2005-2015 was confined to the southern hemisphere with little corresponding heat gain in the northern hemisphere ocean. We propose that this heating pattern of the ocean is driven by anthropogenic climate change and an asymmetric climate variation. This asymmetric variation is found in the pre-industrial control simulations from 11 climate models. While both layers (0-700 m and 700-2000 m) experience steady anthropogenic warming, the 0-700 m layer experiences larger internal variability, which primarily drives the observed hemispheric asymmetry of global ocean heat gain in 0-2000 m layer. We demonstrate that the rate of global ocean warming is consistent with the climate model simulations for this period. Moreover, the observed hemispheric asymmetry in heat gain can be explained by the Earth’s internal climate variability without invoking alternate hypotheses, such as asymmetric aerosol loading.”

Response:- (Abstract, Line 11-21) We appreciate the revisions of the abstract suggested by the reviewer which help us to make it more compelling and also within the word limit of the abstract defined by the journal.

2. Lines 25-27. This sentence is confusing because aerosol forcing of the climate system would lead to loss of energy, not a gain. The gain of energy must be due to positive radiative forcing of the climate system, which as I'm sure the authors know, is dominated by greenhouse gases (and CO₂ in particular).

Response:- We have modified the sentence.

(Introduction, Line 25-27) This energy imbalance is due to the positive radiative forcing of the climate system, which is dominated by the increasing greenhouse gas concentrations, CO₂ in particular⁴.

3. Lines 31-36. It is not clear what this section on the "hiatus" has to do with the rest of the manuscript. Suggest that you either make it clear this is a demonstration of the ability of (unforced?) ocean heat redistribution to modify the observed rates of heat content change or consider removing this paragraph.

Response:- We would like to keep the description of the hiatus and role of internal variability and volcanic forcing in the hiatus. Hence, we have modified the description of the hiatus as suggested by the reviewer.

(Introduction, Line 32-35) This pause in surface warming was the result of trade wind intensification due to the Interdecadal Pacific Oscillation^{12,13}, and with these changes, there was a corresponding redistribution of energy within the oceans¹³⁻¹⁶ and this redistribution is potentially connected to this asymmetric mode.

(Introduction, Line 37-38) The hiatus demonstrates the role of internal climate variability and the natural forcing to modify the observed hemispheric rate of heat content change.

4. Line 37. Remove the word "also".

Response:- We have removed it (Line 39).

5. Line 42. I would challenge the sentence "which have led to a net radiative cooling of the northern hemisphere". Is this really true? This implies that the N Hemisphere *lost* heat during this period (this isn't true of OHC change?). A more robust statement might be "which have contributed to radiative cooling of the northern hemisphere".

Response:- We have modified the sentence as per suggested by the reviewer.

(Introduction, Line 42-45) However, previous studies^{1,2,20} suggest that the asymmetric warming may be related to the natural decadal variability or to the high concentrations of aerosols in the northern hemisphere²¹, which have contributed to the radiative cooling of the northern hemisphere.

6. Line 43. Radiative forcing and internal variability as hypotheses are not mutually exclusive – both may have played a role. In addition, we cannot rule out the possibility that aerosol (or other) forcings may have interacted in some way with internal variability.

Response:- Yes, we agree with the reviewer's point, so we have modified the sentence as per suggested by the reviewer.

(Introduction, Line 42-45) However, previous studies^{1,2,20} suggest that the asymmetric warming may be related to the natural decadal variability or to the high concentrations of aerosols in the northern hemisphere²¹, which have contributed to the radiative cooling of the northern hemisphere.

7. Lines 44-46. "annual to pentadal time scales" seems to contradict "long-term warming". Please clarify this point.

Response: We have modified the statement to avoid the contradiction.

(Introduction, Line 45-48) Moreover, this asymmetric warming is striking in the presence of large-scale increases in the observational records of the ocean temperatures²². This rise in ocean temperature is reflected in the long-term warming of the global ocean, which has shown a rise of OHC in both hemispheres^{22,23}.

8. Lines 46-48. Here and elsewhere. I think that "demonstrate" is too strong a word and implies a greater level of certainty about the real world than we can have based on a small ensemble of climate model simulations. Here's a suggested revised sentence: "Our study shows that the observed asymmetric ocean warming during 2005-2015 can be explained by internal climate variability superimposed on the long-term symmetric anthropogenic ocean warming".

Response: We have modified the statement as per suggestion.

(Introduction, Line 48-50) Our study shows that the observed asymmetric ocean warming during 2005-2015 can be explained by the internal climate variability superimposed on the long-term symmetric anthropogenic ocean warming.

9. Line 51. Replace "robust" with "robustness of the".

Response: (Results, Line 53) We show the robustness of the.....

10. Line 54 / Figure 1. The reader cannot really tell whether the Northern Hemisphere shows a net cooling or not. This requires one to "integrate by eye" over a plot in which the colorscale is saturated. Could you add a line to show the net OHC change, perhaps with a second y-axis?

Response: We agree with the reviewer suggestion. However, we would like to cite Table 1 instead that shows the observed OHC anomaly trend along with the MMM from the average of RCP 4.5 and RCP 8.5 simulation.

(Results, Line 54-57) The depth vs time plot (Fig. 1 and Table 1) of OHC anomaly of the two hemispheres shows the asymmetric character of the upper ocean (0-700 m) with the northern hemisphere cooling and the southern hemisphere warming progressively during 2005-2015.

11. Line 55-58. Here and elsewhere – please avoid subjective terms like "shallower", "shallow", "deep", "long-term" etc. Rather state the depth or time-horizons you are referring to. Based on this

plot, we cannot infer the relative importance of air-sea heat fluxes vs dynamical processes. I think the key point is that we know that a large part of the anti-correlated layers of 0-100m and 100-600m are associated with ocean dynamics in the Tropical Pacific related to ENSO variability. Suggest revising the text to emphasise this point? Final sentence – based on the plot, this variability can be longer than year-to-year (looking at temporal characteristics of the anti-correlated layers)?

Response: We have modified the statement as per the suggestion from the reviewer. We have mentioned the time range and depth range where it is necessary to mention about the long-term and upper/deep ocean.

(Results, Line 57-59) The vertical variations of OHC change in the 0-700 m ocean depth are associated with the ocean dynamics in the Tropical Pacific Ocean related to the El Niño Southern Oscillation (ENSO)^{4,24} variability on interannual or longer time scales.

12. Figure 1 and Figure S1 (and elsewhere?): Please specify the “baseline” period for the Hovmoller plots (i.e. the reference period you compute the OHC anomaly relative to). In both captions – you are not showing “observed ocean heat content”. You are showing “observed ocean heat content anomaly” [heat content would always be positive].

Response: We have modified the caption of Fig. 1 and Supplementary Figure 1 and the text to mention “OHC anomaly” or “OHC change” as per the suggestion from the reviewer.

13. Figure S1. Please format this figure in exactly the same was as Figure 1 in the main manuscript to aid the comparison you are inviting the reader to make in lines 59-64.

Response: We have modified the Supplementary Figure 1, similar to Fig. 1 as per the suggestion from the reviewer.

14. Lines 61-62. Replace “well represented in” with “are similar to”.

Response: We have modified it with the use of “consistent with”.

(Results, Line 62-63) While the observed southern hemisphere trends are consistent with the MMM, this is not the case for the northern hemisphere in 0-700 m.

15. Lines 65-66. Replace “Analysis” with “Inspection” and replace “variability” with “changes”.

Response: We have modified it.

(Results, Line 66-68) Inspection of Fig. 1 suggests that the observed hemispheric asymmetry of the global ocean heat content in the 0-2000 m depth range is predominantly contributed from the changes in the 0-700 m depth range, which is not present in the MMM (Supplementary Figure 1).

16. Lines 67-70. Replace “well represented by” with “consistent with”.

Response: We have modified it.

(Result, Line 68-70) The observed signature of deep ocean warming (700-2000 m) is apparent in both hemispheres and is consistent with the climate model simulations.

17. Section “Linear trend in ocean heat content during 2005-2015”. I think it would be helpful to explain to the reader that while the observed trends represent the combination of internal variability and forced climate change, the MMM will tend to average out internal variability and have a stronger representation of the forced ocean response compared to the observations.

Response: Thank you for making this point; we have used it to enhance the readability and understanding of the reader.

(Results, Line 82-85) While the observed trends represent the combination of internal variability and forced climate change, the MMM will tend to average out internal variability and have a more robust representation of the forced ocean response compared to the observations.

17. Lines 75-77. “.. reduced by 16% during 2005-2015”. Reduced compared to what? Or do you mean that the heat gain is negative and constitutes -16% of the total heat gain in that layer? Please clarify.

Response: We have modified it as suggested by the reviewer.

(Results, Line 76-79) In the 0-700 m layer, the southern hemisphere explains 116% of the net global ocean heat gain, and the northern hemisphere’s rate of ocean heat gain is negative and offsets southern hemisphere contribution by 16% during 2005-2015.

18. Table 1. I think it would be helpful to include the warming rates from the MMM as well. A key point would be the extent to which the 0-2000 m global ocean rates match between models and observations (one could indicate the MMM spread as part of this)?

Response: We have modified the Table 1 as suggested by the reviewer and included the warming rate from the MMM which is the average of RCP 4.5 and RCP 8.5 simulation from the 11 CMIP5 models.

	0-700 m (MMM)	700-2000 m (MMM)	0-2000 m (MMM)
Global Ocean	4.38 ± 0.42 (6.44 ± 1.07)	4.00 ± 0.23 (2.54 ± 0.53)	8.38 ± 0.59 (9.0 ± 1.25)
Southern Hemisphere	5.12 ± 0.45 (3.60 ± 0.67)	2.66 ± 0.19 (1.37 ± 0.39)	7.78 ± 0.58 (5.0 ± 0.79)
Northern Hemisphere	-0.74 ± 0.29 (2.90 ± 0.83)	1.35 ± 0.14 (1.20 ± 0.41)	0.60 ± 0.35 (4.0 ± 1.01)

19. Figure 2. I don’t understand what the methodologies are for the significance testing – this should be made clear somewhere in the manuscript or the supplementary materials. Once the authors have clarified the methods they should make clear what the interpretation of these results for the reader in the main text. An alternative would be to remove the significance testing from the figure. This would allow the authors to focus their discussion on the *differences* in the observed and MMM trends in OHC. This seems to be the key point of the figure? Relative to the MMM, the observations

show higher warming rates in the Southern Ocean and some large areas of cooling in the N. Hemisphere.

Response: We have clearly stated the basis of significance testing in the caption of Fig. 2 (Line 544-550) and also in the Methods section under the subheading Statistical significance.

(Line 544-550) **Fig. 2 Linear temporal trend in ocean heat content anomaly.** (a) Observed Linear trend for 2005-2015 of zonally integrated global ocean heat content anomaly ($1011 \text{ J m}^{-2} \text{ year}^{-1}$) (b) Same as (a) but for MMM trend for 2006-2015, (c) observed linear trend of global ocean heat content anomaly for 0-2000 m ($107 \text{ J m}^{-2} \text{ year}^{-1}$) for 2005-2015, (d) Same as (c) but for MMM trend for 2006-2015. Stippling indicates the locations where OHC anomaly trends are not significant i.e. less than $2 \times$ standard error of the trends estimated from ($n = 6$) observation products and ($n = 11$) CMIP5 models used in this study.

(Statistical significance, Line 358-364) We have used the criterion of $2 \times$ standard error with a sample size of " n " for the significance testing which is equivalent to the 95% confidence from 2-sided student's t-test. The mean trend is significant if it is greater than $2 \times$ standard error of the trends estimated from " n " number of observation and CMIP5 models. Here, " n " represents the sample size which is 6 for observational products and 11 for the CMIP5 model used in this study. The confidence intervals for probability distribution curves are derived from the 2-sigma limits for the gaussian distribution of the random variable that corresponds to a 95% confidence interval from a two-sided student's t-test.

20. Lines 83-93 and elsewhere. I would suggest that the authors use a different term than "well represented" when referring to the MMM simulations compared to the observations, e.g., "consistent" or similar. The thrust of the paper argues that internal variability has played a large role in shaping the observed changes, and we know that the MMM is designed to remove this internal variability – so the idea that observations are "well represented" seems a slightly confusing message.

Response: We have changed the phrase "well represented" to "consistent with" as suggested by the reviewer, whenever it has been used throughout the text.

21. Lines 110-111. Replace "underlying climate change signal" with "externally forced climate change signal".

Response: We have modified it as suggested by the reviewer.

(Results, Line 110-112) The MMM of historical simulations for 0-2000 m (Supplementary Figure 1 and 3) represents the long-term (1980-2015) externally forced climate change signal in which both hemispheres have warmed symmetrically (Supplementary Figure 3a, 3d, 3g).

22. Lines 116-117. Replace "symmetric warming patterns" with "similar warming rates".

Response: We have modified it as suggested by the reviewer.

(Results, Line 115-117) Similarly, the MMM of the RCP 4.5 and RCP 8.5 simulations show similar warming rates in both hemispheres (Supplementary Figure 3).

23. Line 118-119. Please cite a figure and make it clear that you are referring to a spatially more uniform pattern of warming. I believe the authors are referring to Figure S2b.

Response: We have cited the figure as suggested by the reviewer.

(Results, Line 118-120) The observed OHC anomaly trend in the 700-2000 m depth layer is not obscured by the upper ocean internal variability and represents the long-term warming signature across the globe (Supplementary Figure 2b) and in both hemispheres (Fig. 1).

24. Lines 124-128 and Figure S4. To help the reader, please introduce a final panel to this figure that shows the MMM result (there is a space for this bottom right).

Response: We have modified the Supplementary Figure S4 with the addition of MMM panel "l" as suggested by the reviewer.

25. Line 137-139. Replace "analysis shows" with "analysis suggests" and insert "externally forced" before "anthropogenic warming".

Response: We have modified that statement.

(Results, Line 138-140) This analysis suggests that the observed hemispheric asymmetry of 0-2000 m in the global OHC change (Fig. 1 and Fig. 2a, 2c) is a combination of the internal variability and externally forced anthropogenic warming.

26. Line 141. Please replace "from climate change" with "from forced response" or similar. Climate change can arise through both internal variability or external forcings, especially given the relatively short time-horizons being discussed in this paper.

Response: We have modified it.

(Results, Line 142) Separating internal variability from the forced response

27. Line 142. Replace "the worlds without" with "representations of the climate system in the absence of".

Response: We have modified it.

(Results, Line 143-145) Using the CMIP5 Pi-Ctrl simulations (Supplementary Table 1), which represent the climate system in the absence of anthropogenic forcing, we investigate whether the observed trend in OHC anomaly during the 11-year Argo period is consistent with internal variability alone.

28. Line 145. Remove the word "event".

Response: We have modified it.

(Results, Line 145-146) The distribution of internal variability in the Earth system is commonly used in attribution studies⁴.

29. Line 161-162. Replace “the Earth’s total energy due to changes in the net radiation balance” with “global ocean heat content that implies corresponding changes in the net top-of-atmosphere radiation balance”.

Response: We have modified it.

(Results, Line 160-163) It is worth mentioning that there are some instances when the internal variability has in-phase components of heat in both hemispheres and corresponds to the changes in global ocean heat content that implies corresponding changes in the net top-of-atmosphere radiation balance²⁸.

30. Line 167. In relation to comment above – either change to “top-of-atmosphere” here or amend comment above. Either way, best to use consistent naming convention throughout the manuscript.

Response: We have modified it.

(Results, Line 167-169) We also consider that the 0-2000 m depth represents the full water column that responds to the internal variations in the net top-of-atmosphere radiation balance^{9,28} with the minimum role of vertical ocean heat exchanges.

31. Figure 3. The caption is very long. I suggest the authors try to reduce this by putting some of this information in the methods section.

Response: We have modified the caption of Fig. 3 and transferred some of the material to the method section.

(Methods , Line 347-356) The Monte-Carlo simulations of 10-year periods from all 11 models were then concatenated into a single series to generate an MME to represent the distribution of all OHC trends due to internal variability (cloud in Fig. 3a). The critical thing to note is that the same 10-year period from the Monte-Carlo simulations was used to estimate trends in the global, northern, and southern hemisphere analyses. Furthermore, to represent the climate change due to external forcing, we have shifted the cloud of internal variability (green cloud in Fig. 3b, 3c, 3d and 3e) by the average of the trend estimated from RCP 4.5 and RCP 8.5 i.e. $\left(\frac{RCP4.5_{MMM} + RCP8.5_{MMM}}{2}\right)_{2006-2015}$ which is shown by the orange cloud in Fig. 3b, 3c, 3d and 3e. The trajectory of historical observation (Hist-Obs 1980-2016, Fig. 3) is computed from the 10 year running trends from the long-term observations over the period of 1980-2016 with a sliding window of 12 months.

32. Lines 271-272. Suggest hyphenating “signal-to-noise”, here and elsewhere.

Response: We have modified it.

(Results, Line 92-93) This suggests that the ocean below 700 m holds the key to tracking Earth’s warming due to climate change since the signal-to-noise ratio there is much higher.

(Discussion, Line 272-273) Monitoring the deep ocean has distinct advantages for tracking climate change because of weaker internal variability leading to a much higher signal-to-noise ratio.

33. Line 278-280. “Entirely due” is a categorical statement, which is unnecessary, and open to criticism. Suggest re-phrase along the lines of “model analysis shows that the observed net heat gain by the global ocean is driven by anthropogenic external forcing of the climate system and that the hemispheric asymmetry in warming rates can be explained by internal climate variability”.

Response: We have modified it.

(Discussion, Line 280-282) Our analysis shows that the observed net heat gain by the global ocean is driven by anthropogenic external forcing of the climate system and that the internal climate variability can explain the hemispheric asymmetry in warming rates.

34. Line 281-283. Suggest the text is replaced with: “This phenomenon requires substantial changes in net hemispheric air-sea heat exchanges and/or cross-equatorial net ocean heat transport”. This statement of course assumes a minimal role for vertical ocean heat exchange – it may be worth stating this somewhere explicitly in the paper (perhaps here?) since I believe it is a central assumption to the analysis?

Response: We have modified it.

(Discussion, Line 284-285) This phenomenon requires substantial changes in net hemispheric air-sea heat exchanges¹⁹ and/or cross-equatorial net ocean heat transport^{30,32}.

We have also mentioned the assumption of vertical ocean heat exchanges in Line 166-168

(Results, Line 167-169) We also consider that the 0-2000 m depth represents the full water column that responds to the internal variations in the net top-of-atmosphere radiation balance^{9,28} with the minimum role of vertical ocean heat exchanges.

35. Line 288. Again, I question the evidence for “net radiative cooling” – see comment 5.

Response: We have modified it.

(Discussion, Line 289-293) Despite the high concentration of aerosols in the northern hemisphere that have contributed to its radiative cooling^{1,2,20}, the combination of the anthropogenic warming and the internal variability of the climate system provides a sufficient and likely explanation for the anomalously enhanced (reduced) rate of ocean heat gain in the southern (northern) hemisphere during 2005-2015.

36. Method section. Equation (1) and (2) appear to be identical? Please remove the second occurrence.

Response: We have modified it.

(Methods, Line 335-338) After de-drifting the models, we computed OHC from 11 CMIP5 models used in this study by using equation (1) with constant seawater density of (1025 kg m^{-3}) ^{28,29,44} along

with the ocean potential temperature (θ) and C_p of $3992 \text{ J kg}^{-1} \text{ K}^{-1}$. We then computed the MMM trend from historical, RCP 4.5, and RCP 8.5 simulations.

37. Lines 315-318. This sentence is not clear. Are you saying that the models were chosen because they have a better representation of the S. Ocean mean state? (i.e. relatively smaller biases?). I think you can probably merge these sentences.

Response: We have modified it.

(Methods, Line 316-319) The choice of these 11 CMIP5 models is based on a previous study³⁰ which shows that the Southern Ocean has high heat content between 40° - 50° S and 11 (out of 12) CMIP5 models robustly capture the pattern of high heat storage on the northern flank of the Antarctic Circumpolar Current, i.e. zonal band of 40° - 50° S.

38. Line 324. "climate forcing" is not observed. I would simply state that the forcings and experimental design follow the CMIP5 protocol. Please include references to CMIP5 and the experimental documentation here (I believe that is Taylor et al, 2012 for CMIP5, and Meinshausen et al, 2011, for RCPs. I'm not sure what papers describe "historical" and "pre-industrial" experiments).

Response: We have modified it.

(Methods, Line 322-324) From these CMIP5 models, we used Pi-Ctrl, historical simulations (Hist-CMIP5, 1980-2005), RCP 4.5, and RCP 8.5 simulations (2006-2015 and 2020-2100) which follow the forcing and experimental design from the CMIP5 protocol²⁷.